# Analysis of rod/cone gap junctions from the reconstruction of mouse photoreceptor terminals

**Munenori Ishibashi[1], Joyce Keung[1], Catherine W Morgans[2], Sue A Aicher[2], James R Carroll[2], Joshua H Singer[3], Li Jia[4], Wei Li[4], Iris Fahrenfort[1], Christophe P Ribelayga[1]\*†, Stephen C Massey[1]\***

[1]Richard Ruiz Department of Ophthalmology and Visual Science, McGovern Medical School, University of Texas at Houston, Houston, United States; [2]Department of Chemical Physiology & Biochemistry, Oregon Health & Science University, Portland, United States; [3]Department of Biology, University of Maryland, College Park, College Park, United States; [4]Retinal Neurophysiology Section, National Eye Institute, National Institutes of Health, Bethesda, United States

**\*For correspondence:**
cpribela@central.uh.edu (CPR);
steve.massey@uth.tmc.edu
(SCM)

**Present address:** †Department of Vision Sciences, University of Houston College of Optometry, Houston, United States

**Competing interest:** The authors declare that no competing interests exist.

**Abstract** Electrical coupling, mediated by gap junctions, contributes to signal averaging, synchronization, and noise reduction in neuronal circuits. In addition, gap junctions may also provide alternative neuronal pathways. However, because they are small and especially difficult to image, gap junctions are often ignored in large-scale 3D reconstructions. Here, we reconstruct gap junctions between photoreceptors in the mouse retina using serial blockface-scanning electron microscopy, focused ion beam-scanning electron microscopy, and confocal microscopy for the gap junction protein Cx36. An exuberant spray of fine telodendria extends from each cone pedicle (including blue cones) to contact 40–50 nearby rod spherules at sites of Cx36 labeling, with approximately 50 Cx36 clusters per cone pedicle and 2–3 per rod spherule. We were unable to detect rod/rod or cone/cone coupling. Thus, rod/cone coupling accounts for nearly all gap junctions between photoreceptors. We estimate a mean of 86 Cx36 channels per rod/cone pair, which may provide a maximum conductance of ~1200 pS, if all gap junction channels were open. This is comparable to the maximum conductance previously measured between rod/cone pairs in the presence of a dopamine antagonist to activate Cx36, suggesting that the open probability of gap junction channels can approach 100% under certain conditions.

## Editor's evaluation

This article presents a beautiful analysis of gap junctions in the outer retina using a combination of confocal imaging and electron microscopy. The result is a thorough description of connectivity between rod and cone photoreceptors, and a clear resolution of ambiguities present in past work on this topic.

## Introduction

Signaling between neurons is served by both chemical synapses and electrical connections known as gap junctions. Chemical transmission is the dominant mode, but electrical synapses can change the properties and routing of local networks and microcircuits (*Marder et al., 2017*). Gap junctions contain clusters of intercellular channels that allow the passage of ions and other small molecules, and so support electrical coupling. At a gap junction, the membranes of two neighboring cells are closely

**eLife digest** Neurons can talk to each other in two ways: they can send chemical messengers across specialized junctions between two cells, or they can directly pass electrical signals to one another. This latter process is made possible by gap junctions, a system of channel-like structures which connect neighbouring cells and let ions move between them. In most neurons, gap junction channels are made from a specialized protein called connexin 36. Gap junctions are small, difficult to observe, and therefore often ignored by researchers studying neural circuits.

In response, Ishibashi et al. focused on nerve cells in the mouse retina, in particular the cones (which detect color during the day) and the rods (which are essential for night vision). Gap junctions between rods and cones allow them to communicate; for example, they enable rod signals to directly activate cones. This provides an alternative route for rod signaling known as the 'secondary rod pathway', which seems to be open at night and switches to closed around dawn.

Both rods and cones only produce connexin 36, so Ishibashi et al. labeled these proteins with fluorescent tags to pinpoint gap junctions. This showed that each cone makes around 50 gap junctions with nearby rods; however, gap junctions were not detected between cells of the same type.

In addition, 3D reconstruction helped to establish the length of each gap junction. Further experiments showed that a typical rod was connected to a cone by about 80 connexin 36 channels. Finally, calculations revealed that the gap junction channels would all need to open to account for the level of electrical activity required for the secondary rod pathway. This suggests that gap junctions may be much more active and important than previously thought.

The work by Ishibashi et al. provides a new understanding of the number, size and activity of gap junctions in the retina, potentially paving the way to prevent diseases where light-sensing cells degenerate and cause blindness.

apposed or 'zippered,' leaving only a 2–4 nm gap (*Bloomfield and Völgyi, 2009*; *Miller and Pereda, 2017*). Each gap junction channel is formed of two docked hemichannels, or connexons, and each connexon is assembled from six connexin subunits. Connexin expression is required on both sides to form a gap junction (*Miller et al., 2017*; *Jin et al., 2020*). The vertebrate connexin family in mouse includes 20 members, including Cx36, the most common neuronal connexin (*Beyer and Berthoud, 2009*; *Söhl et al., 2004*).

The pattern of gap junction connectivity between neurons confers distinct properties on circuit function. Homologous gap junctions between cells of the same type support the lateral spread of signals through coupled networks that can average over a wider area than a single neuron, as would be expected for rod/rod or cone/cone coupling, if they were shown to occur. In addition, heterologous coupling between different cell types can provide the specific connections to form another neuronal pathway or an alternative route for signal flow. For example, rod/cone coupling provides an alternative pathway for rod signals to enter cone-driven circuits, known as the secondary rod pathway. The gap junction connections between AII amacrine cells and ON bipolar cells in the retina, which support the primary rod pathway, are another well-known example of heterologous coupling (*Demb and Singer, 2012*; *Feigenspan et al., 2001*; *Mills et al., 2001*; *Veruki and Hartveit, 2002*).

While the distribution of gap junctions is widespread in neural tissue and their roles in signal averaging, noise reduction, synchronization, and predictive coding have been well documented (*Connors, 2017*; *Nagy et al., 2018*), they are small, difficult to image, and often ignored. Thus, the role of electrical coupling in neural circuits may be underappreciated and poorly understood. Indeed, it is challenging to reliably identify gap junctions in electron microscopy (EM) material (*Pallotto et al., 2015*), and gap junctions may be absent from large-scale serial EM reconstructions (*Kasthuri et al., 2015*; *Scheffer et al., 2020*). In this study, we demonstrate an approach for measuring small gap junctions reliably in the mouse retina that can be applied to other parts of the brain.

In the retina, the synaptic endings of photoreceptors, known as cone pedicles and rod spherules, terminate in the outer plexiform layer (OPL) where they make synapses with second-order neurons: horizontal cells and bipolar cells. Cone pedicles are found in a single layer in the mid-OPL, whereas the smaller rod spherules are found above and between the cone pedicles in the top (distal) half of the OPL. In the mouse retina, rods outnumber cones by about 30-1 (*Carter-Dawson and LaVail, 1979*),

and, numerically, the OPL is dominated by rod spherules. Cone pedicles are the largest structures in the OPL. They contain numerous synaptic ribbons, each marking an active zone, and make synaptic contacts with horizontal cells and 13 types of cone bipolar cell. One of these bipolar cell types, CBC9, contacts short wavelength-sensitive (i.e., 'blue') cones selectively, which can be identified on this basis (*Behrens et al., 2016*; *Nadal-Nicolás et al., 2020*). Rod spherules contain a single large ribbon and synapse with horizontal cells and a single type of rod bipolar cell.

The numerous, small gap junctions in the OPL were first described in early ultrastructural studies in cat and primate retinas (*Kolb, 1977*; *Raviola and Gilula, 1973*; *Smith et al., 1986*). Basal processes, called telodendria, spread laterally from cone pedicles to make small gap junctions with rod spherules. In freeze-fracture EM, the gap junctions appear as strings of single particles curved around the synaptic invagination of rod spherules (*Raviola and Gilula, 1973*). The presence of gap junctions at rod/cone contacts is consistent with the electrical transmission of rod signals to cones, forming the secondary rod pathway in which rod signals influence cone-driven circuits (*Asteriti et al., 2014*; *Ingram et al., 2019*; *Li et al., 2010*; *Nelson, 1977*; *Ribelayga et al., 2008*; *Ribelayga and Mangel, 2010*; *Schneeweis and Schnapf, 1995*).

Rods and cones both express Cx36, but no other connexins (*Bloomfield and Völgyi, 2009*; *Jin et al., 2020*; *O'Brien et al., 2012*) with Cx36-mediated rod/cone coupling accounting for most of the gap junctions in the OPL (*Asteriti et al., 2017*; *Ingram et al., 2019*; *Jin et al., 2020*). In addition to rod/cone gap junctions, cone/cone coupling has been reported in cat and primate retinae (*Kolb, 1977*; *Smith et al., 1986*). Rod/rod gap junctions have been suggested from tracer coupling studies and EM studies of the mouse retina (*Jin et al., 2015*; *Li et al., 2012*; *Tsukamoto et al., 2001*), and, in salamander retina, rods are extensively electrically coupled (*Zhang and Wu, 2004*). However, the evidence for rod/rod gap junctions in the mammalian retina is mixed (*Bloomfield and Völgyi, 2009*; *Bolte et al., 2016*; *Jin et al., 2020*; *Tsukamoto et al., 2001*) and will be addressed here.

In this study, we used a combination of confocal microscopy, serial blockface-scanning electron microscopy (SBF-SEM) and focused ion beam-SEM (FIB-SEM), to address four major goals: (i) reconstruct the cone telodendrial network, (ii) identify all (both homologous and heterologous) gap junction types in the OPL, (iii) resolve the question of rod/rod coupling, and (iv) make a quantitative estimate of rod/cone coupling.

We used a publicly available SBF-SEM dataset (e2006; *Helmstaedter et al., 2013*), which contains photoreceptor terminals and the OPL, to map membrane contacts between photoreceptors. Unfortunately, because this dataset was prepared to enhance membrane contrast and facilitate tracing neuronal processes, gap junctions were not identifiable. To work around this problem, we used confocal microscopy to identify the location of Cx36 gap junctions on rod spherules (*Jin et al., 2020*). Tracing the contacts between cone telodendria and rod spherules in the SBF-SEM dataset revealed numerous potential sites that corresponded with the location of Cx36 immunofluorescence in the confocal dataset. Thus, the combination of confocal microscopy and SBF-SEM is complementary. To confirm the location of rod/cone gap junctions, we used FIB-SEM on new samples processed to preserve ultrastructure, which allowed high-resolution imaging of contacts between cone pedicles and rod spherules.

We confirmed that heterologous rod/cone gap junctions account for the vast majority of gap junctions in the OPL (*Jin et al., 2020*). The combination of imaging methods enabled us to determine the pattern of connectivity between photoreceptors and make a quantitative estimate of rod/cone coupling, based on gap junction size and number. Finally, we correlated this morphological data with our previously published measurements of gap junction conductance between photoreceptors obtained from paired rod/cone recordings (*Jin et al., 2020*). This combination of morphological and physiological data provides a way to estimate some of the fundamental properties of gap junctions and how they may contribute to a neuronal circuit. Our calculations suggest that most connexon channels, up to 100% under certain conditions, can contribute to the wide dynamic range of these small, string-like gap junctions between rods and cones.

## Results

### Localization of Cx36 in the OPL: Confocal microscopy

Immunofluorescent labeling for Cx36 revealed small clusters of labeling in the OPL (*Figure 1A*, red). To locate these potential gap junctions, we used an antibody to the vesicular glutamate transporter (vGlut1) (*Figure 1B*, blue) to identify rod spherules and an antibody to cone arrestin (CAR) to label cone pedicles (*Figure 1C*, green). Labeling with the CAR antibody stains cones in their entirety from the outer segment to the pedicle (*Figures 1C and 2A*). Fine processes, called telodendria (e.g., oblique arrows in *Figure 1C*), extend laterally from each cone pedicle to form an overlapping matrix in the OPL. The area between and above the cone pedicles is filled with numerous vGlut1-labeled rod spherules while the space underneath is occupied by the dendrites of horizontal cells and bipolar cells.

The cone telodendria rose above the level of the cone pedicles to contact the overlying rod spherules (*Figure 1C*). In the OPL, there are many small Cx36 clusters associated with cone telodendria, often at their tips, relatively high (distal) in the OPL. Staining the rod spherules for a synaptic vesicle marker, such as the vesicular glutamate transporter, shows that Cx36 clusters are contained within the band of vGlut1-labeled rod spherules (*Figure 1B*, *Figure 1—video 1*). There was almost no Cx36 labeling in the outer nuclear layer (ONL). This is important because it rules out the presence of Cx36 gap junctions between rod somas and/or passing axons in the ONL, where they are packed together at high density. Cx36 clusters are also apparent distinctly underneath each cone pedicle (vertical arrowheads, *Figure 1*), on processes previously identified as bipolar cell dendrites (*Feigenspan et al., 2004*; *O'Brien et al., 2012*; *Raviola and Gilula, 1973*). Because they are not colocalized with cone pedicles, they are excluded from further analysis in this article.

### Number of Cx36 clusters on individual cones

When all the cones are labeled for CAR, the complexity of the overlapping matrix makes it difficult to analyze individual cone pedicles with confidence. Therefore, we turned to a transgenic mouse line, *Opn4^cre;Z/EG* (*Ecker et al., 2010*), where there is sparse labeling of a few individual cones (*Figure 2A*). This made it possible to view individual EGFP-labeled cones against a background of all cones stained for CAR (*Figure 2B*). The immunofluorescence data were analyzed by extracting only the Cx36 clusters that colocalized with EGFP-labeled pedicles (*Figure 2C*) and reconstructing in 3D to find the number and location of Cx36 clusters on a single-cone pedicle (*Figure 2D*, *Figure 2—video 1*). At each Cx36 cluster, there is an adjacent rod spherule (*Figure 2E*), suggesting that these structures are rod/cone gap junctions. There are approximately 51.4 ± 8.88 (mean ± SD, n = 18, three retinae) Cx36 clusters on each cone pedicle (*Figure 2D and F*, *Figure 2—source data 1*), along the telodendria and over the upper surface of the cone pedicle.

### Localization of Cx36 clusters on blue cone pedicles

Blue cones initiate a specific color-coded pathway in the mammalian retina (*Behrens et al., 2016*; *Dacey and Lee, 1994*; *Haverkamp et al., 2005*; *Kouyama and Marshak, 1992*). Blue cones in the dorsal retina were stained in their entirety by use of a blue cone opsin Venus transgenic mouse line (*Figure 2—figure supplement 2*). Alternatively, blue cones were identified using an antibody to blue cone opsin to stain the outer segment and then following the CAR-labeled axon down to the pedicle in a confocal series (*Figure 2—figure supplement 3*, *Figure 2—video 2*). In either case, a sparse mosaic of 'true blue' cones was located in the dorsal retina, as opposed to the ventral retina where most cones express both blue (S) and green (M) opsins (*Nadal-Nicolás et al., 2020*). We find that the pedicles of blue cones have telodendria-bearing Cx36-labeled clusters in a manner indistinguishable from green cone pedicles (*Figure 2G*, *Figure 2—figure supplement 3*). Thus, it appears that blue cones also make numerous Cx36 gap junctions with rods.

### Cx36 clusters are located in all rod spherules, close to the opening of the postsynaptic compartment

In high-resolution confocal images (×63 objective, NA 1.4, Zeiss Airyscan), the Cx36 elements occur exclusively where cone telodendria contact the base of each rod spherule (*Figure 3A*). Some rod/cone contacts have multiple Cx36 clusters (*Figure 3A*, arrow). In these images, rod spherules appear as oval structures (blue) including two unlabeled compartments or holes, separated by the synaptic

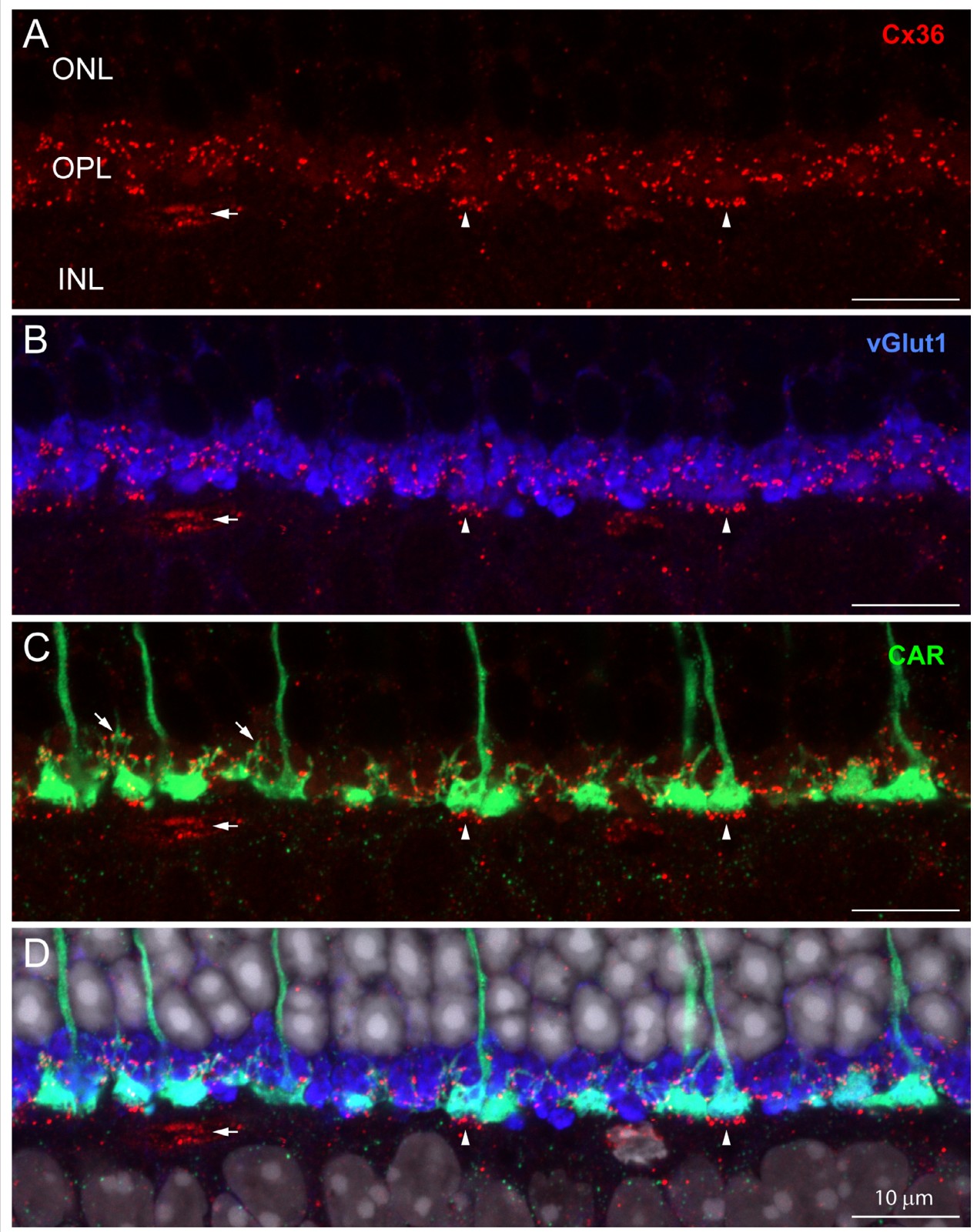

**Figure 1.** The distribution of Cx36 in the outer plexiform layer (OPL). Cx36 labeling in the OPL, confocal microscopy. (**A**) Numerous small Cx36 clusters (red) restricted to the OPL, absent in the outer nuclear layer (ONL). For all four panels: horizontal arrowheads, bipolar cell Cx36 clusters under each cone pedicle that were excluded from analysis because they were not colocalized with cone pedicles; horizontal arrow, non-pecifically labeled blood vessel. (**B**) Cx36 clusters are contained within the band of rod spherules, stained with an antibody against vGlut1 (blue). (**C**) Cx36 clusters decorate the cone

*Figure 1 continued on next page*

*Figure 1 continued*

pedicles and their telodendria, labeled for cone arrestin (green). Oblique arrows point to examples of cone telodendria. (**D**) Four channels showing Cx36 (red), cone pedicles (green), and rod spherules (blue) all contained within the OPL. DAPI-labeled nuclei (gray) show the well-organized ONL. *Figure 1—video 1* shows this dataset.

The online version of this article includes the following video for figure 1:

**Figure 1—video 1.** Animation of a confocal series from the outer plexiform layer (OPL) showing Cx36 (red) decorating the cone telodendria (green, cone arrestin).

https://elifesciences.org/articles/73039/figures#fig1video1

---

ribbon, immunolabeled for ribeye (*Figure 3B*, white). The lower hole in the rod spherule of *Figure 3B* contains the mGluR6-labeled (green) tips of rod bipolar cell dendrites and was thus identified as the postsynaptic compartment. The upper hole contains a single large mitochondrion, labeled for the mitochondrial translocase TOMM20 (red). To quantitatively determine the position of Cx36-labeled points, we captured vGlut1-labeled rod spherules and generated a mean structure by aligning, stacking, and averaging the images from 18 complete rod spherules from a single section. The sites of individual Cx36 elements were marked and found to be located around the base of the mean rod spherule (*Figure 3C and D*). A spline curve was fitted to the outline of the mean rod spherule and, after linearizing this curve, the density of Cx36 was plotted. The twin peaks of the resulting curve show that Cx36 (red) is distributed around the mouth of the synaptic opening, shown by a drop in the vGlut1 labeling (blue) (*Figure 3E*). In the same region, the labeling for the cone signal (green) is high, suggesting that Cx36 gap junctions occur at telodendrial contact points with rods. Two potential outliers, at approximately 3 o'clock and 9 o'clock (*Figure 3C*), were actually located on other nearby rod spherules and so were excluded from the analysis. In this sample of 18 rod spherules, there were 45 Cx36 clusters, yielding 2.50 ± 0.764 Cx36 clusters/rod (mean ± SD). From the mean structure of these 18 rod spherules, all Cx36 elements were located at the base of the rod spherule, within 1–2 µm of the opening to the postsynaptic compartment (*Figure 3E*).

To estimate the fraction of rods that are coupled to cones, we generated a larger sample, including the 18 rods above. We counted the number of Cx36-labeled points at the base of each rod spherule where there was contact with a cone, excluding any rod spherules that were not completely contained within the section. From seven different sections, from three retinae, all 260 rod spherules analyzed have cone contacts and Cx36 labeling close to the synaptic opening (*Jin et al., 2020*). Frequently, there are multiple Cx36-labeled points at the base of a single rod spherule; the number of Cx36 elements ranged from 1 to 6 with a mean of 2.48 ± 1.01 (mean ± SD) (*Figure 3F and G*, *Figure 3—source data 1*). Presuming that a Cx36 cluster indicates the presence of a gap junction, we conclude that every rod spherule was coupled to a nearby cone.

## Distribution of photoreceptors in the OPL: Serial blockface electron microscopy

The e2006 SBF-SEM dataset is derived from a block of mouse retina including 164 cone pedicles and thousands of rod spherules with a voxel size of 16.5 × 16.5 × 25 nm (*Helmstaedter et al., 2013*). Cone pedicles are easily recognized as the largest structures in the OPL, and we were able to map them and register the resulting map with the data from *Behrens et al., 2016* (*Figure 4A*). From these data, we could locate the blue cone pedicles previously identified by their selective contacts with blue cone (CBC9) bipolar cells (*Behrens et al., 2016*). Rod spherules were also easily identified as the numerous round and compact structures with prominent postsynaptic inclusions in the synaptic invagination. In this dataset, rod spherules are 2–3 µm in diameter and they are massed above and surrounding the cone pedicles (*Figure 4B*), usually oriented with the synaptic invagination at the base.

## Skeletons show many contacts between cone telodendria and rod spherules

To assess potential gap junctional contacts between cone pedicles and rod spherules, we chose a patch of 29 adjacent cones (*Figure 4A*, box 1) to skeletonize, meaning we followed their processes to nearby contacts or termination. This patch included 13 central cone pedicles within the ring surrounded by an annulus of 16 additional cone pedicles (*Figure 4A and C*). Some calculations were

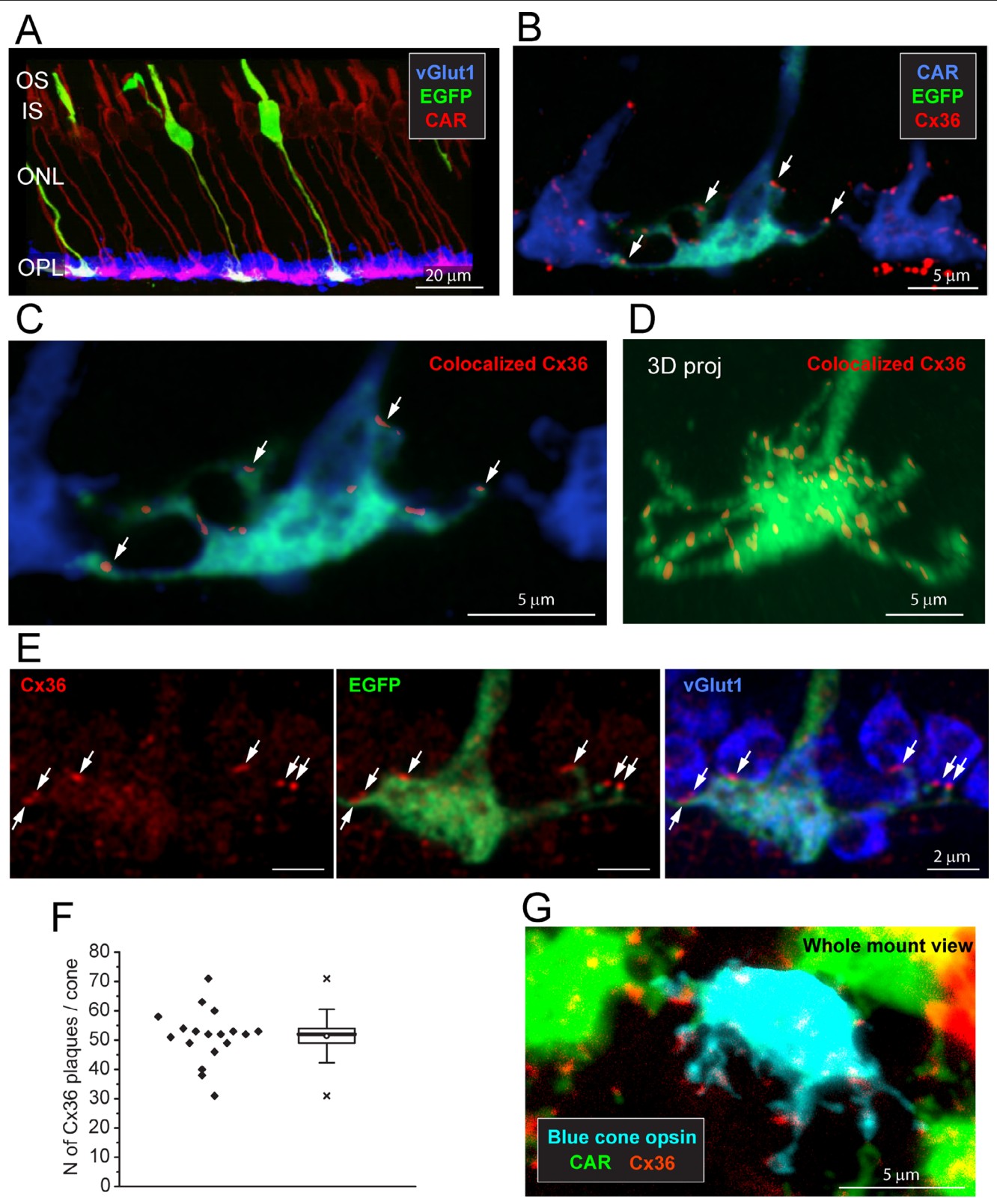

**Figure 2.** Cx36 is colocalized with cone pedicles. (**A**) EGFP-labeled single cones (green) against a background of all cones stained for cone arrestin (red) and vGlut1, which stains rod spherules (blue). Most cone pedicles appear magenta because they are labeled for both cone arrestin and vGlut1. The EGFP-labeled cone pedicles are white, triple labeled for EGFP, cone arrestin, and vGlut1. (**B**) A mini-stack of five optical sections (0.5 μm in total) showing the distribution of Cx36 (red) on cone pedicles (blue), including a single EGFP-labeled cone (green + blue = cyan). Arrows point to individual

*Figure 2 continued on next page*

*Figure 2 continued*

Cx36 clusters. Images of individual channels are available in *Figure 2—figure supplement 1A*. (**C**) Enlarged version, same mini-stack showing Cx36 clusters colocalized with the single EGFP-labeled cone pedicle. Note the similarity with panel (**B**) because Cx36 is predominantly colocalized with cone pedicles and there is very little nonspecific Cx36 background. Additional images of the same size as (**B**) are available in *Figure 2—figure supplement 1B*. (**D**) 3D projection of a complete single EGFP cone pedicle (green) compiled from confocal sections showing only Cx36 clusters (red) colocalized with this cone pedicle, as in (**C**). From such data, we calculated the mean number of Cx36 clusters per cone pedicle (**F**, below). Images of individual channels are available in *Figure 2—figure supplement 1C*. *Figure 2—video 1* shows this dataset. (**E**) A single confocal section showing that rod spherules (blue) occur close to the Cx36 clusters (red) located on a cone pedicle (green). Left: Cx36 clusters (red), arrows point to individual Cx36 clusters (all panels). There is some faint background labeling for Cx36, which is mostly contained within the cone pedicle. Center: the distribution of Cx36 clusters (red) on a cone pedicle (green) and its telodendria. Right: triple label showing that rod spherules, labeled for vGlut1 (blue), contact the cone pedicle (green +blue = cyan) at Cx36 clusters (red). (**F**) The number of Cx36 clusters, 51.4 ± 8.88 (mean ± SD), on a single reconstructed cone pedicle (as in **D**), box shows quartiles, mean (circle), median (bold line), SD (whisker), min/max (x), n = 18. (**G**) Blue cone pedicle (cyan), identified in a blue cone opsin Venus mouse line, had telodendria-bearing Cx36 clusters (red), similar to those of green cones, stained for cone arrestin (green).

The online version of this article includes the following video, source data, and figure supplement(s) for figure 2:

**Source data 1.** Number of Cx36 clusters on a single-cone pedicle.

**Figure supplement 1.** 3D reconstruction of a single EGFP-labeled cone pedicle and colocalized Cx36 clusters.

**Figure supplement 2.** The distribution of blue cone opsin in the mouse retina.

**Figure supplement 3.** Blue cone pedicles have telodendria-bearing Cx36 clusters.

**Figure 2—video 1.** 3D reconstruction of a single EGFP-labeled cone pedicle among all cone arrestin (blue)-labeled cones, with extensive telodendria.
https://elifesciences.org/articles/73039/figures#fig2video1

**Figure 2—video 2.** Animating from the outer plexiform layer (OPL) to the outer segments following the cone axon allows cone labeled for blue cone opsin (magenta) to be connected to its pedicle (*), labeled for cone arrestin.
https://elifesciences.org/articles/73039/figures#fig2video2

based on the central 13 cones to avoid edge artifacts. All skeleton data are provided in *Figure 4—source data 1* and summarized in *Appendix 2—table 1*. This central area contained one blue cone pedicle, but, in addition, we analyzed all the blue cone pedicles in the dataset (*Figure 4A*), as identified by *Behrens et al., 2016* (summarized in *Appendix 2—table 2*). We also examined several other locations, including an area with a sparse distribution of cone pedicles (*Figure 4A*, box 2), to be sure we did not select an atypical area (*Figure 5—figure supplement 1C and D*). The mosaic of cone pedicles with their overlapping telodendria is plotted in *Figure 4C*. Cone axons were followed into the ONL, and a projection shows that the telodendria are contained within the upper part of the OPL (*Figure 4D*).

We started our analysis with a single-cone pedicle (cone 5), near the center, which had a complex but typical field of telodendria approximately 10 μm in diameter, and we marked the position of every rod spherule in contact with this cone (*Figure 5A*). Cone 5 has contacts with 34 rod spherules that included every rod spherule within the telodendrial field. In a projection, it can be seen that the telodendria extend laterally and above the cone pedicle, reaching up to the overlying rod spherules (*Figure 5A*). Many, but not all, telodendria terminate at the base of a rod spherule. The skeleton of a blue cone pedicle (cone 2) is similar (*Figure 5A, bottom, B and C*, arrow).

Outlining the telodendrial fields and adding the rest of the cone pedicles shows a dense matrix of overlapping telodendria (*Figure 5B*) each with an area of 104 ± 0.2 μm² (mean ± SD, n = 29, *Figure 5F*, *Figure 5—source data 1*) and a coverage estimated at 1.56 (n = 29 cones). Each cone contacts 43.0 ± 5.40 rod spherules (mean ± SD, n = 29 cones) (*Figure 5G*, *Figure 5—source data 1*), including every rod spherule within its field, typically at the base. Plotting convergence (number of rod spherule contacts per cone) against the pedicle area shows a slight trend for the largest cone pedicles to contact more rods spherules (*Figure 5H*, *Figure 5—source data 1*).

Examining these contacts from the perspective of the rod spherules, we coded the overlying rod spherules in the same color as each cone pedicle. Rod spherules with contacts from two or three of the central 13 cone pedicles are marked black or dark gray, respectively, with noncontacted rod spherules marked as light gray. Viewing the resulting map quickly makes the point that most rods are contacted by two or three cone pedicles (*Figure 5C*, *Figure 5—figure supplement 1A*). Omitting the cone pedicles (except for one example) and all rod spherules in contact with those cones leaves only an annulus of rod spherules outside the field of reconstructed cones and demonstrates the almost total absence of rods with no cone contacts (*Figure 5—figure supplement 1B*). We found 3/811 (0.4%) rods without

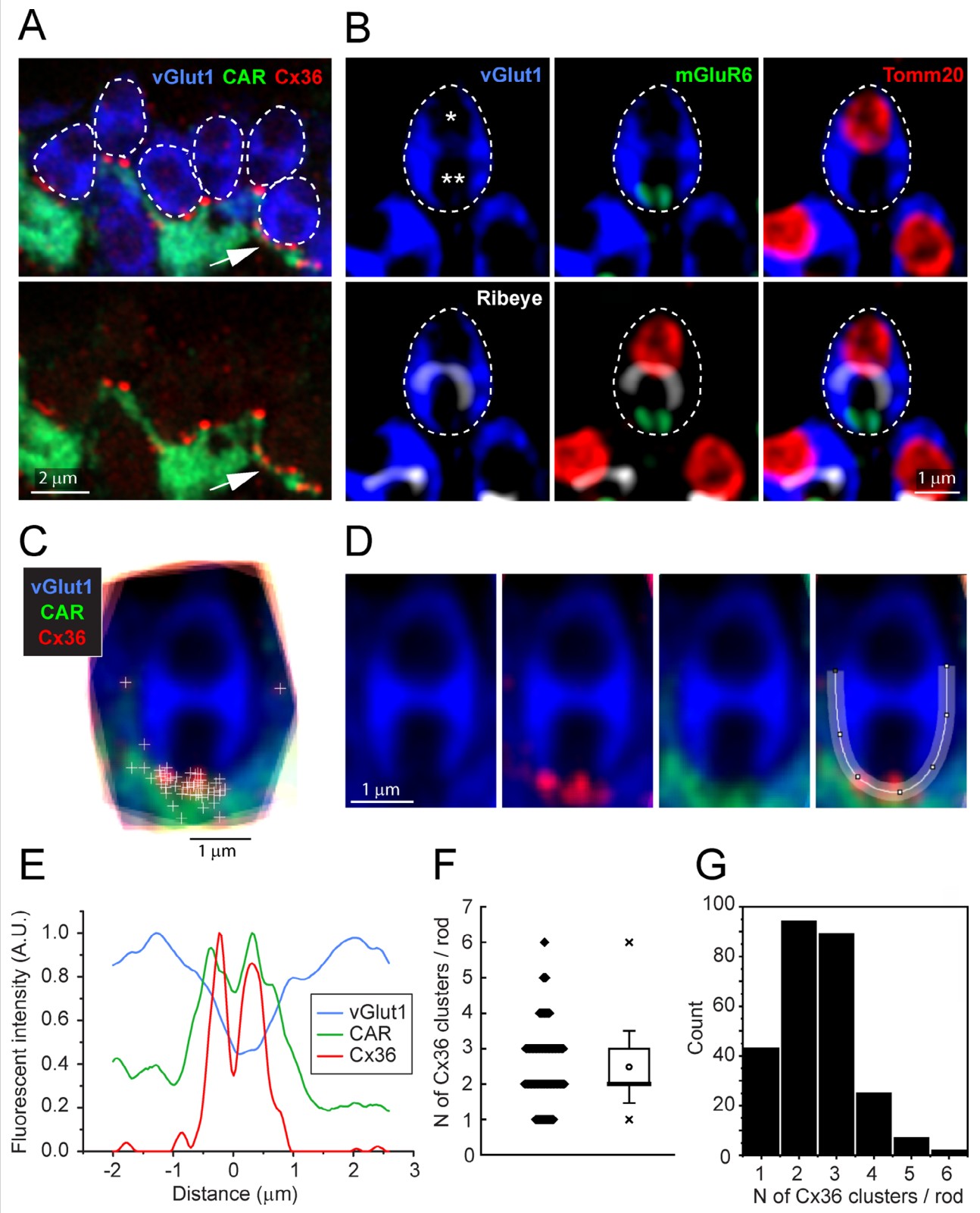

**Figure 3.** Cx36 clusters are found at the base of each rod spherule.

(**A**) Top: details of outer plexiform layer (OPL), rod spherules, stained for vGlut1 (blue) and outlined with dashed lines are located above and between cone pedicles labeled for cone arrestin (green). Cx36 clusters (red) are found at the base of rod spherules where they contact cones. Multiple Cx36 clusters occur at some contacts (arrow). Bottom: Cx36 clusters are located on cone telodendria. Images of individual channels are available in

*Figure 3 continued on next page*

*Figure 3 continued*

*Figure 3—figure supplement 1A*. (**B**) Top left: a rod spherule labeled for vGlut1 (blue), outlined with a dashed line, has two compartments (* and **). Top middle: bottom compartment is open at the base and contains two mGluR6-labeled rod bipolar dendrites, identifying the postsynaptic compartment. Top right: the upper compartment is filled with a large mitochondrion, labeled for TOMM20 (red). Lower left: the synaptic ribbon, labeled for ribeye (white), lies between the two compartments, arching over the postsynaptic compartment. Lower middle: the synaptic ribbon lies between the mitochondrion and the mGluR6 label. Bottom right: all four labels superimposed showing a typical rod spherule. Images of individual channels with visible background are available in *Figure 3—figure supplement 1B*. (**C**) 18 rod spherules, each from one optical section, aligned and superimposed. Taking the rod spherule as a clock face (top center is 12 o'clock), Cx36 clusters (red with individual puncta marked +), are found at the base, 6 o'clock, along with cone telodendria (green). The two single +s at 3 o'clock and 9 o'clock are associated with other rod spherules and were excluded from the analysis. (**D**) Average rod spherule showing Cx36 and cone telodendria at the base with a fitted spline curve. (**E**) The distribution of label in each confocal channel along a linearized version of the spline curve. Where the vGlut1 label is low, indicating the opening to the postsynaptic compartment, the Cx36 (red) and cone (green) signals are high. The double peak for Cx36 indicates clusters on each side of the synaptic opening. (**F**) Scatterplot showing the number of Cx36 clusters, presumed to be gap junctions, per rod spherule from a sample of 260 rods, mean 2.48 ± 1.01, box shows quartiles, mean (circle), median (bold line), SD (whisker), min/max (x). (**G**) Data plotted as histogram showing the distribution of multiple Cx36 clusters per rod spherule, n = 260 rods, 645 gap junction points.

The online version of this article includes the following source data and figure supplement(s) for figure 3:

**Source data 1.** Number of Cx36 clusters on a rod spherule.

**Figure supplement 1.** Location of Cx36 and the fine structure of rod spherules.

cone contacts within the field of 29 cones, an insignificant fraction. As a precaution, we also checked a sparse patch with a rare hole in the cone mosaic (*Figure 5—figure supplement 1C and D*). Even here, more than 95% of rods received cone contacts.

If each cone contacts every rod spherule within its telodendrial field, then in areas where the cone telodendria overlap, each rod must receive contacts from multiple cones. This is clearly the case, and, in fact, most rods receive contacts from two, three, or, rarely, four cones. A simplified map for two adjacent cones is shown in *Figure 5D*, while the contact map for all 29 cones is shown in *Figure 5—figure supplement 2A*. Of approximately 40 rods in contact with each cone pedicle, only a few have exclusive contacts with a single cone (7.23 ± 3.38, mean ± SD, n = 13 central cone pedicles). Most rods receive contacts from several cone pedicles while adjacent cone pedicles share as many as 23 rod spherules, (mean ± SD = 6.23 ± 4.67, n = 79 cone pairs) (*Figure 5—figure supplement 2B*, *Figure 5—source data 2*). There is a pronounced edge effect because the outermost ring of cone pedicles lacks the overlap from further unanalyzed cones: to avoid this, we analyzed the rod contacts of the central 13 cone pedicles. In this area, rods with multiple cone contacts were the norm; 74% of rod spherules received contacts from two or three cones, yielding a rod to cone divergence of 1.89 (*Figure 5E*). This reflects the density of the telodendrial network and the large number of rod/cone gap junctions in the OPL. Clearly, each rod spherule has the potential to make several contacts with close-by cones, in agreement with the confocal data showing the distribution of Cx36.

## Blue cone skeletons also contact rod spherules

The skeleton of a blue cone, defined by its bipolar cell contacts (*Behrens et al., 2016*; *Nadal-Nicolás et al., 2020*), is also shown in *Figure 5A* and the analysis of all five blue cones from the dataset of *Behrens et al., 2016* is presented for comparison with green cones (*Figure 5F and G*). The blue cone data are summarized in *Appendix 2—table 2*. The telodendrial area of blue cones was 85.1 ± 18.6 μm$^2$ (mean ± SD, n = 5) (*Figure 5F*). Like green cones, blue cones contact all rod spherules, within their telodendrial field (convergence 39.8 ± 5.11, mean ± SD, n = 5). The convergence in blue cones fell within the range of rods in contact with green cones, 43.0 ± 5.40 rod spherules (mean ± SD, n = 29). Most of the rods in contact with a blue cone also receive contact from adjacent overlapping green cones. Thus, there is no evidence for color-selective rod contacts.

## 3D reconstruction of cone pedicles to visualize contacts with rod spherules

We segmented several cone pedicles and the rod spherules contacted by each cone pedicle. The contact sites, where the cell membrane of a cone telodendron combined with the cell membrane of a rod spherule, with no visible space between them, were highlighted (*Figure 6A–D*). These contact pads could then be superimposed on either the cone pedicle or the surface of the overlying rod

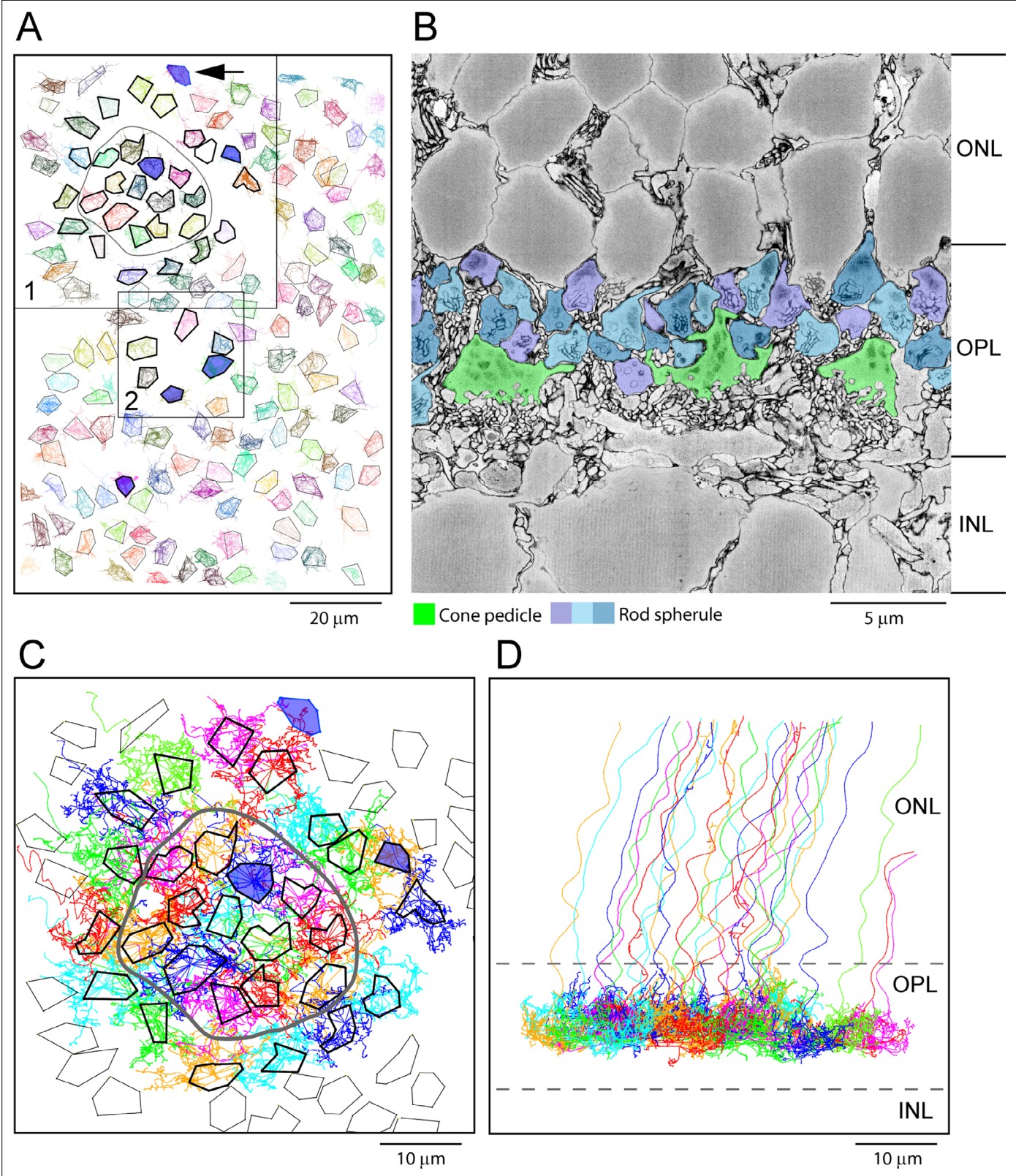

**Figure 4.** Map of cone pedicles shows telodendria overlap.

(**A**) Map of 164 cone pedicles in e2006, modified from *Behrens et al., 2016*. Box 1 contains the 29 cone pedicles (thick outlines) that were reconstructed as skeletons in (**C**). The central 13 cone pedicles are contained within the ring. Blue cone pedicles filled with dark blue. Box 2 is a lower density area where six additional cone pedicles were also reconstructed as a check (*Figure 5—figure supplement 1C*). In total, there were six blue cone

*Figure 4 continued on next page*

*Figure 4 continued*

pedicles, filled dark blue (*Behrens et al., 2016*). One blue cone pedicle was at the upper edge (black arrow). The cone pedicles outside the boxes were not reconstructed, except for the single blue cone pedicle that lies outside the boxes. (**B**) Low-power vertical section from e2006 showing rod spherules (three shades of blue), above and in between cone pedicles (green). (**C**) Skeletons of all 29 reconstructed cone pedicles (thick outlines) showing overlapping telodendrial fields, individually colored. Thick outlines show solid part of cone pedicles. The central 13 cone pedicles are contained within the ring. (**D**) Projection of (**C**) showing cone pedicles restricted to the outer plexiform layer (OPL) and axons ascending through the outer nuclear layer (ONL).

The online version of this article includes the following source data for figure 4:

**Source data 1.** Cone pedicle skeletons and rod spherule points.

---

spherules. Typically, the telodendrial contacts, often from more than one cone, appear as arcs, close to the synaptic opening of rod spherules (*Kolb, 1977*; *Smith et al., 1986*; *Tsukamoto et al., 2001*). Confocal analysis also demonstrated multiple cone contacts with a single rod spherule (*Figure 6E*).

The reconstruction of adjacent cones 3 and 5 shows a complex field of telodendria extending laterally and upward (distally) from the pedicles (*Figure 6F*). Their telodendria interdigitate, mostly avoiding each other; the sparse cone-to-cone contacts will set a limit on cone/cone coupling. We also reconstructed all the rod spherules in contact with these cones (*Figure 6G*, *Figure 6—video 1*, *Figure 6—video 2*). Most of the overlying rod spherules have telodendrial contacts (*Figure 6A and C*) but a few make direct contacts with the upper surface or roof of the cone pedicle (*Figure 6B and D*, *Figure 6—figure supplement 1A*). When the contact sites with rods are displayed on the cone pedicles, the appearance is similar to the 3D reconstruction of confocal material showing the distribution of Cx36 clusters on a single-cone pedicle (*Figures 2D and 6H*). When the contact sites are displayed on the rod spherule lower surfaces, it is obvious that most telodendrial contacts are very close to the mouth of the synaptic invagination, often forming a curved line or horseshoe around it (*Figure 6I and J*), within 1–2 µm. Neighboring cone pedicles often contact the same rod spherules. At such sites, the contact pads from both cone pedicles can be found, often interspersed around the synaptic mouths of the rod spherule (*Figure 6J and K*, *Figure 6—video 1*, *Figure 6—video 2*). The cone contact sites are consistent with the location of Cx36 near the mouth of the synaptic invagination and thus indicate the potential presence of rod/cone gap junctions (*Figure 6E*). However, a contact pad may exceed the extent of a gap junction and therefore does not predict its size (see below).

Of the cone pedicles that were completely reconstructed, cones 5, 3, and 2 (a blue cone) had 57, 59, and 54 contact pads, respectively. Thus, the number of contact pads is close to the number of Cx36 clusters per cone pedicle (51.4 ± 8.88, mean ± SD), from the confocal analysis, and this suggests that most of the contact pads include gap junction sites. There were a few examples where a cone telodendron contacts a rod spherule far from the synaptic invagination, often in transit to another location, and these can show large areas of contact. But these sites are not associated with Cx36 labeling in the confocal data. In other words, these are incidental contacts of passage, not gap junctions, and they were excluded from our analysis.

There are a few anomalies in the OPL: some rod spherules sit directly astride a cone pedicle and receive few or no telodendrial contacts. Instead, they make direct contacts with the roof of the cone pedicle (*Figure 6—figure supplement 1*). In addition, the lowest (most proximal) row of rods in the ONL do not have axons or spherules (*Li et al., 2016*). Instead, the synaptic machinery is included in the lowest crescent of the cell body, adjacent to the OPL. These low rods represent the most distal synaptic structures in the OPL, yet they still receive contacts from cone telodendria (*Figure 6—figure supplement 2*). Finally, there are a few rod spherules below (proximal to) the level of the cone pedicles. These are often inverted so the mouth is on top, and this is the site of telodendrial contacts (*Figure 6—figure supplement 3*). Thus, in every case, including these anatomical variations, cone contacts occur close to the rod synaptic opening, coincident with the location of Cx36. These sites very probably contain Cx36 rod/cone gap junctions.

## 3D reconstruction of a blue cone pedicle also shows contacts with rod spherules

Based on the work of *Behrens et al., 2016*, who identified the cones in contact with the blue cone bipolar cell (CBC9), we could locate the blue cones in the map of cone pedicles. The telodendrial field of cone 2, a blue cone, contacts 41 rod spherules and overlaps substantially with the surrounding cones

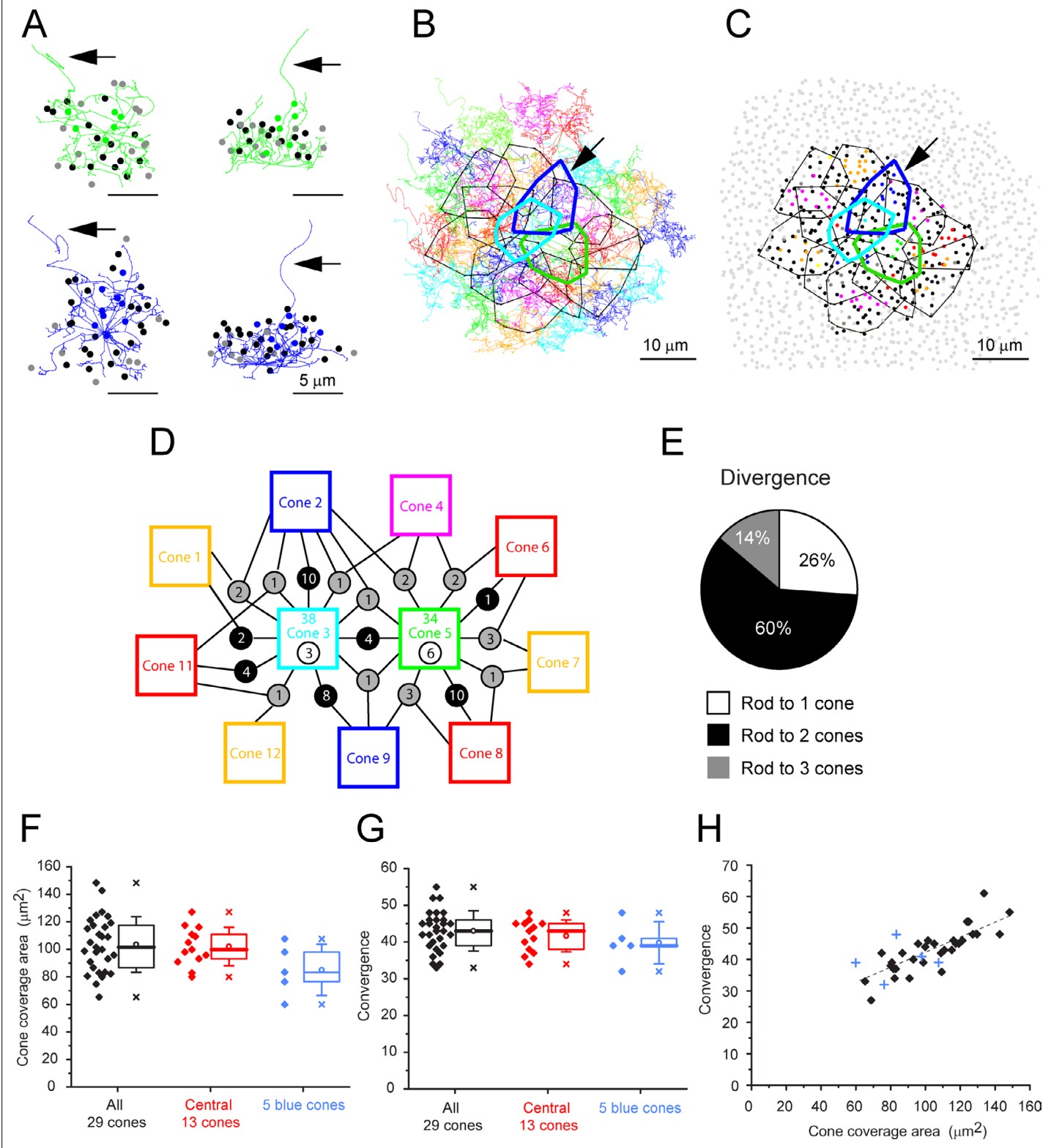

**Figure 5.** Cone skeleton analysis shows cones contact all nearby rod spherules.

(**A**) Skeletons of one green cone (cone 5, green) and one blue cone (cone 2, blue) in wholemount view and projected. Black arrows show ascending axons. The position of each contacted rod spherule is marked by a dot, color-coded the same color (green or blue) if the contacts are exclusive to this cone pedicle; black, contacts two cones; gray, contacts three cones. (**B**) Telodendrial fields of 29 reconstructed cone pedicles, each color-coded; central 13 outlined by polygons, arrow points to blue cone (cone 2, thick blue outline). Cones 3 and 5 are also outlined with cyan and green, respectively.

*Figure 5 continued on next page*

*Figure 5 continued*

(**C**) Outlines of central 13 cone pedicles showing all rod spherule contacts, color-coded, exclusive to one cone; black, contacts with two cones; dark gray, contacts with three cones. Light gray, rod spherules outside the range of the central 13 cone pedicles. Arrow points to blue cone (cone 2, thick blue outline). (**D**) Simplified skeleton map showing contacts of two green cone pedicles (cones 3 and 5); the box contains the cone identity, the total number of rod contacts (above), and the exclusive number of rods that contact this cone only (below). Circles indicate the number of rod spherules shared between cones linked by lines to neighbors. Cone 2 is a blue cone, which shares rod contacts with neighboring cones. (**E**) Pie chart showing distribution of cone contacts per rod spherule. (**F**) Area of telodendrial field for all 29 reconstructed cone pedicles, central 13 cones (ring in *Figure 4A and C*) and 5 blue cones. Box shows quartiles, mean (circle), median (bold line), SD (whisker), min/max (x). (**G**) Convergence, rod contacts per cone for all 29 reconstructed cone pedicles, central 13 (ring in *Figure 4A and C*) and five blue cones. Box shows quartiles, mean (circle), median (bold line), SD (whisker), min/max (x). (**H**) Convergence vs. cone pedicle area. Line of linear regression shows some tendency for larger cone pedicles to contact more rods. +, blue cones.

The online version of this article includes the following source data and figure supplement(s) for figure 5:

**Source data 1.** Cone coverage area and rod convergence.

**Source data 2.** The number of rod spherules shared between two adjacent cone pairs.

**Figure supplement 1.** Nearly all rods receive cone contacts.

**Figure supplement 2.** Rod connectivity map for 29 cone pedicles.

---

(*Figure 6—figure supplement 4A*). This blue cone pedicle made the same types of rod contacts as other cones, including telodendrial contacts, roof contacts, and contacts with the rods in the bottom row of the ONL (*Figure 6—figure supplement 4B*). Reconstructing in 3D shows typical contact pads around the mouth of the overlying rod spherules with alternating contributions from multiple cones, including this blue cone (*Figure 6—figure supplement 5*). Thus, blue cone pedicles also make rod/cone gap junctions and both blue and green cones can make gap junctions with the same rod. We found no evidence of color-specific coupling.

## SBF-SEM dataset shows electron-dense rod/cone gap junctions

While the SBF-SEM data analyzed above provided extensive information about close appositions between cone pedicles and rod spherules, the sample preparation was not appropriate to resolve definitive gap junctions in the tissue. To demonstrate that the contact pads described above represent the presence of gap junctions, we also examined rod/cone contacts in an SBF-SEM dataset (eel001) in which synapses and rod/cone gap junctions were visible. In *Figure 7A and B*, a roof contact between a cone pedicle and an overlying rod spherule is shown. There is clear separation between the cell membranes until a darkly stained chromophilic area where the membranes merge, indicating a gap junction approximately 350 nm in length. The postsynaptic inclusions of the rod spherule show that this site is close to the opening of the postsynaptic compartment. In a brief survey of the SBF-SEM data, we found 13 examples of darkly stained merged membranes at rod/cone contacts, and a much larger number in the FIB-SEM dataset (see below). We have shown above that Cx36 occurs at such contacts between rods and cones and therefore conclude that this confirms the presence of a gap junction.

## Size and distribution of rod/cone gap junctions revealed by FIB-SEM

FIB-SEM provides isotropic data (same resolution in each dimension), facilitating 3D reconstruction (*Xu et al., 2017*), and this allows measurement of gap junction dimensions. Thus, we also obtained two FIB-SEM datasets of mouse OPL (FIB-SEM 1 and FIB-SEM 2) with 4 nm voxels revealing electron-dense staining consistent with the size and position of Cx36 labeling, which we interpret as gap junctions. Contacts from cone telodendria at the base of each rod spherule show darkly stained gap junctions (*Figure 7C and D*) in the same position as Cx36 labeling determined by confocal microscopy (*Figure 3*). It should be noted that the area of telodendrial contact is frequently larger than the size of the gap junction (*Figure 7D*). Thus, while contact pads in the SBF-SEM e2006 images indicate the potential for a gap junction (*Figure 6H–K*), they do not predict its size.

Sometimes, what appear as small separate gap junctions in a single section merge in nearby sections to form one large continuous gap junction. An example is shown in *Figure 7E–H*; two small gap junctions on either side of the synaptic opening are part of a horseshoe structure similar in shape to many of the reconstructed contact pads. Reconstructing in 3D shows one large gap junction encircling the

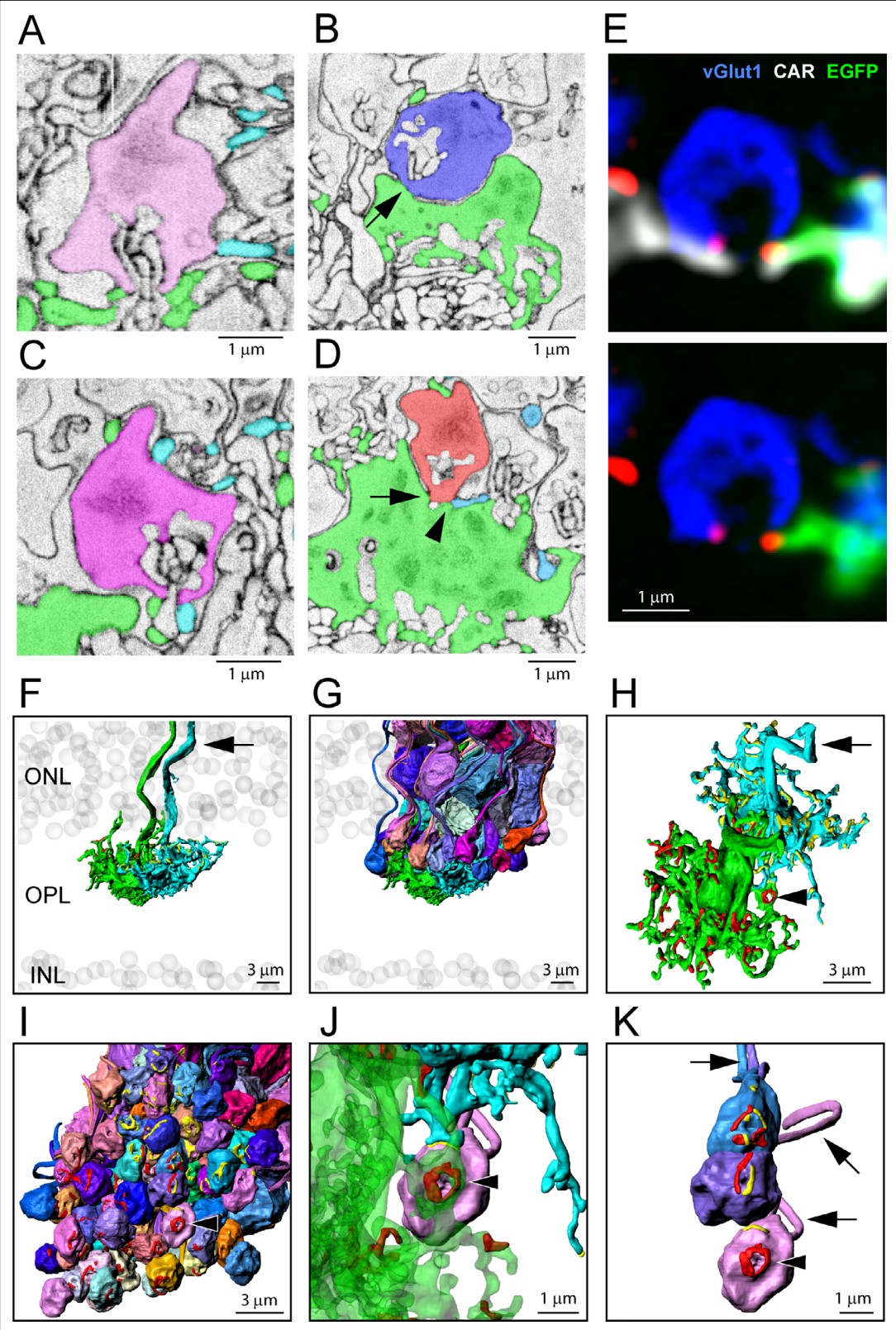

**Figure 6.** Segmentation and 3D reconstruction e2006: cone telodendria contact rod spherules close to the mouth of the synaptic opening.
 (**A**) Single serial blockface-scanning electron microscopy (SBF-SEM) section, telodendria from two cones (green or cyan), rod spherule (pink), in the plane of the synaptic opening with cone contacts on both sides. (**B**) Single SBF-SEM section, rod spherule (blue), postsynaptic inclusions show it is close to the synaptic opening, 25 nm × 4 sections = 100 nm away. The rod spherule sits on the roof of a cone pedicle (green) with a contact, shown

*Figure 6 continued on next page*

*Figure 6 continued*

by the arrow, where membranes merge. (**C**) Single SBF-SEM section, rod spherule (magenta) with contacts at the synaptic opening from two different cones (green or cyan). (**D**) Single SBF-SEM section, rod spherule (orange), contacts (arrow) the roof of a cone pedicle (green) with a telodendrial contact (arrowhead) from a second cone (cyan). Postsynaptic inclusions in the rod spherule indicate the contacts are close to the synaptic opening, 25 nm × 9 sections = 225 nm away. (**E**) Top: confocal microscopy, single optical section, shows Cx36 clusters (red) where telodendria from two different cones (white for cone arrestin, labels both cones; green for a single EGFP-labeled cone) contact rod spherule (blue), at the mouth of the synaptic opening. Bottom: cone arrestin signal turned off for clarity, showing only one cone is labeled for EGFP. (**F**) 3D reconstruction from e2006, two adjacent green cone pedicles (cone 5, green, and cone 3, cyan), ghost cell bodies (gray) show limits of outer nuclear layer (ONL) and inner nuclear layer (INL), arrow marks ascending cone axons. (**G**) Same two cone pedicles with all 66 ( = 38 + 34–6 shared) reconstructed rod spherules contacted by these two cones. (**H**) Rotated view, looking down at top surface of same two cone pedicles, contact pads with rod spherules marked in red or yellow for each cone. Arrowhead marks a single rod spherule; arrow ascending axon. (**I**) Rotated view, looking up at the bottom surface of all rod spherules in contact with these two cones, contact pads in red or yellow, arrowhead marks a single rod spherule magnified in panels (**J**) and (**K**). (**J**) Single rod spherule (pink), green cone pedicle (cone 5), with adjusted transparency, with contact pad (red) encircling the synaptic opening at the base of the rod spherule (arrowhead). The second cone (cone 3, cyan) also contacts this rod spherule nearby (yellow), approximately 1 µm from the synaptic mouth. (**K**) Detail showing the bottom surface of three adjacent rod spherules that receive contacts close to the synaptic opening from both cone pedicles, contact pads in red or yellow, arrows show rod axons, arrowhead indicates same rod spherule as (**J**). *Figure 6—video 2* shows this dataset.

The online version of this article includes the following video and figure supplement(s) for figure 6:

**Figure supplement 1.** Confocal microscopy, roof contacts.

**Figure supplement 2.** Low rods.

**Figure supplement 3.** Low rod spherules.

**Figure supplement 4.** Blue cones, segmentation, e2006.

**Figure supplement 5.** Blue cones e2006, 3D reconstruction.

**Figure 6—video 1.** 3D reconstruction of a single cone (cone 5, green) with contact pads in red, then all rod spherules that were in contact with this cone.

https://elifesciences.org/articles/73039/figures#fig6video1

**Figure 6—video 2.** A pair of reconstructed adjacent cone pedicles (cone 5, green, and cone 3, cyan).

https://elifesciences.org/articles/73039/figures#fig6video2

base of the rod spherule with a length of approximately 1.5 µm (*Figure 7H*, *Figure 7—video 1*). This curved structure is highly reminiscent of the curved gap junction strings revealed by freeze-fracture EM (*Raviola and Gilula, 1973*). In the OPL, a few large gap junctions of a similar shape, >1 µm long, were readily apparent in the confocal view of wholemount retina labeled for Cx36 (*Jin et al., 2020*).

We were able to rotate these gap junction contacts in 3D and estimate the gap junction size from the en face view (*Figure 7—figure supplement 1*), as well as the distance from the mouth of the postsynaptic compartment. We analyzed 42 complete rod spherules with a total of 135 rod/cone gap junctions. These data are summarized in *Appendix 2—table 3*. In the two FIB-SEM datasets, all 42 rod spherules analyzed have gap junctions close to the mouth of the synaptic compartment consistent with the location of Cx36 (*Figure 3A–E*). We could not always trace the contact back due to the small volume of these high-resolution datasets, but when it was possible, in 112 cases, 100% of the gap junction contacts were identified as a cone. The number of gap junctions per rod spherule ranged from 1 to 6 with a mean of 3.21 ± 1.23 (mean ± SD, n = 135, *Figure 8A*, *Figure 8—source data 1*). The mean distance from the center of synaptic mouth was 0.686 ± 0.635 µm, (mean ± SD, n = 135), but this is a skewed distribution. 84% of gap junctions lay within 1 µm of the synaptic mouth with a median distance of 0.435 µm; 94% were within 2 µm (*Figure 8B and C*, *Figure 8—source data 1*), coincident with the position of cone contacts and the location of Cx36 labeling (*Figure 3C–E*). There were eight gap junctions, 6% of the total, which fell outside this range. When these outliers were traced back, the contacts were identified as cones in every case, so despite their distant location relative to the rod synaptic mouth, they were identified as rod/cone gap junctions.

The identity of the dense chromophilic material visible in EM pictures of gap junctions is unknown, but it may consist of gap junction proteins (connexins) in addition to scaffolding proteins and modulatory subunits that are thought to make up a gap junction complex (*Lasseigne et al., 2021*; *Nagy et al., 2018*). Freeze-fracture EM analysis of retinal gap junctions has shown that they exist in several distinct forms, from plaques with a crystalline array to sparse linear forms such as ribbons and strings (*Kamasawa et al., 2006*). En face, the rod/cone gap junctions had a relatively uniform width (~120 nm), suggesting a standardized component with a variable length such as a string, as previously reported

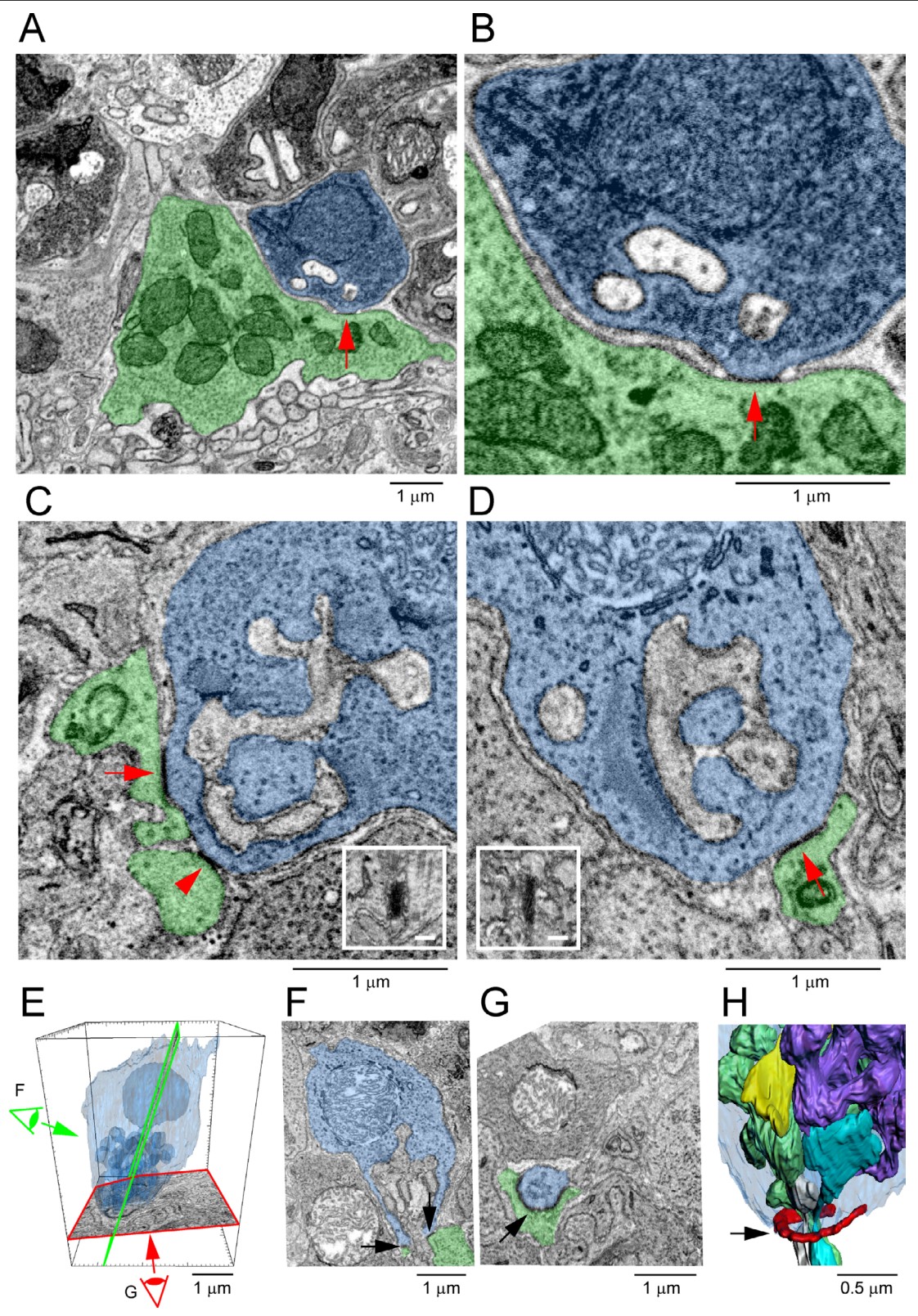

**Figure 7.** Segmentation and 3D reconstruction show gap junctions at rod/cone contacts.
 (**A**) Serial blockface-scanning electron microscopy (SBF-SEM) dataset (eel001), a rod spherule (blue) nestled on the roof of a cone pedicle (green) with a densely stained gap junction contact (red arrow). (**B**) Magnified image of (**A**), detail showing an electron-dense gap junction (red arrow) with merged membranes. Note that there is clear separation except at this point. Postsynaptic inclusions indicate the gap junction is close to the synaptic opening

*Figure 7 continued on next page*

*Figure 7 continued*

of the rod spherule. (**C**) Focused ion beam-scanning electron microscopy (FIB-SEM): electron-dense gap junction staining at contact points (red arrow and arrowhead) between cone telodendria (green) and a rod spherule (blue). Inset: rotated en face view of one gap junction (red arrow) to measure length. (**D**) FIB-SEM: second example (red arrow) of an electron-dense gap junction at a cone contact (green) with a rod spherule (blue). Inset: en face view of the gap junction. Note that green cone telodendrial contact exceeds the gap junction staining. (**E**) FIB-SEM: overview of a rod spherule (blue) with a transparent reconstruction over the image planes in (**F**) and (**G**). Note the large mitochondrion towards the top of the rod spherule. (**F**) Image plane of the rod spherule (blue) through the synaptic opening with a small gap junction on each side (arrows) at a cone contact (green). (**G**) Image plane close to the base of the rod spherule (blue) reveals the two small gap junctions in (**F**) are part of a single large gap junction. There is a horseshoe-shaped gap junction (arrow) close to the synaptic opening where a cone telodendron (green) wraps around the base of the rod spherule. Large Cx36 horseshoe-shaped Cx36 clusters like this are easily observed by confocal microscopy. (**H**) FIB-SEM 3D reconstruction of the rod spherule (transparent blue) showing a single large gap junction (red, arrow) around the synaptic opening at the base. Postsynaptic inclusions are also rendered with two rod bipolar dendrites (gray and cyan) and two horizontal cell processes (green and purple). The synaptic ribbon is shown in yellow, partially obscured by the postsynaptic processes. *Figure 7—video 1* shows this dataset.

The online version of this article includes the following video and figure supplement(s) for figure 7:

**Figure supplement 1.** Measurement of gap junction (GJ) size in focused ion beam-scanning electron microscopy (FIB-SEM).

**Figure 7—video 1.** 3D reconstruction showing a large horseshoe-shaped gap junction (red) around the base of a transparent rod spherule with postsynaptic processes also filled (from focused ion beam-scanning electron microscopy [FIB-SEM]).
https://elifesciences.org/articles/73039/figures#fig7video1

---

in freeze-fracture studies of the OPL (*Raviola and Gilula, 1973*). This is also consistent with the low fluorescent intensity of Cx36 in the OPL compared to the IPL, where gap junctions occur predominantly in the form of bright Cx36 clusters representing dense two-dimensional arrays of connexons (*Kamasawa et al., 2006*).

The mean length of a rod/cone gap junction was 477 ± 227 nm (mean ± SD, n = 135, 42 rods, *Figure 8D*, *Figure 8—source data 1*). *Raviola and Gilula, 1973* reported the photoreceptor gap junctions as strings of single particles with a spacing of 10 nm. Therefore, we estimate that the average rod/cone gap junction contains a string of 48 connexons (477/10 ≅ 48). The total gap junction length per rod spherule was 1.53 µm ± 0.439 µm (mean ± SD, n = 42, *Figure 8E*, *Figure 8—source data 1*) or 150 channels. The largest individual gap junctions occurred when there was only one gap junction per rod spherule (*Figure 8D, F, and G*, red points, *Figure 8—source data 1*); these were outliers with a length close to the mean total length and they were visible in confocal images as large curved Cx36-labeled structures (*Jin et al., 2020*).

In summary, the quantitative analysis of rod/cone gap junctions from the FIB-SEM data confirms the presence of gap junctions at the sites identified from the contact analysis of SBF-EM dataset e2006 and the confocal analysis of Cx36 labeling. Furthermore, rod/cone gap junctions appear as concentric strings around the postsynaptic opening, as reported in the original freeze-fracture data (*Raviola and Gilula, 1973*; *Reale et al., 1978*).

## Cone/cone gap junctions were not detected

To evaluate cone/cone coupling, we segmented and partially reconstructed six cone pedicles from dataset FIB-SEM 1. The small size/high resolution of the dataset meant that no cone pedicles were complete. Nevertheless, we found 22 contacts between six pairs of cones, 3–4 contacts per pair (*Figure 9A*). This suggests that a single-cone pedicle could make around 20 contacts with up to six surrounding pedicles given a telodendrial coverage of 1.5.

Although we were able to locate cone/cone contacts, they did not have the typical appearance of a gap junction. While the contacts were direct, without intervening glial processes, there was no clear membrane density. When a nearby rod/cone gap junction was contained in the same frame, it was much more densely stained (*Figure 9B*). We examined all 22 cone/cone contacts, but we were unable to identify any typical electron-dense gap junctions in this material. This is surprising given previous reports of cone/cone coupling in mammals.

## Exclusion of rod/rod gap junctions

Previous work has reported the presence of rod/rod gap junctions (*Tsukamoto et al., 2001*). However, in our FIB-SEM sample of 42 reconstructed rod spherules, we were unable to locate any rod/rod gap junctions. The location and packing density of rod spherules means they are often adjacent, but we

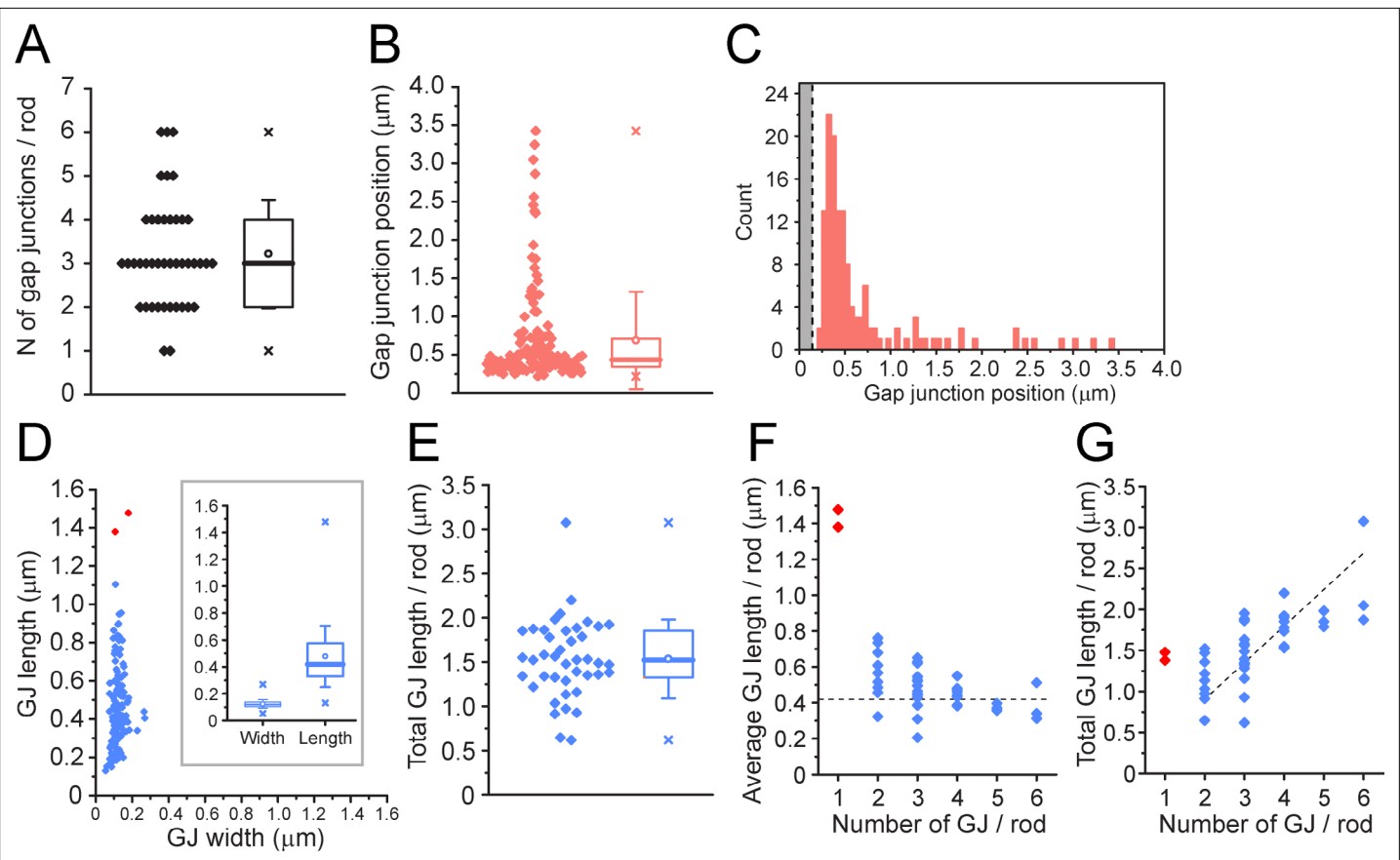

**Figure 8.** Quantitative analysis of 42 reconstructed rod spherules from focused ion beam-scanning electron microscopy (FIB-SEM). (**A**) The number of gap junctions per rod spherule ranged from 1 to 6 (3.21 ± 1.23, mean ± SD, n = 42). Box shows quartiles, mean (circle), median (bold line), SD (whisker), min/max (x). (**B**) Gap junction distance from the rod spherule synaptic opening, skewed distribution, median, 0.435 µm. Box shows quartiles, mean (circle), median (bold line), SD (whisker), min/max (x). (**C**) Histogram plots of gap junction position, gray area shows radius of the synaptic opening (0.138 ± 0.121 µm, n = 42 rod spherules). (**D**) Gap junction length vs. width for 135 gap junctions from 42 rod spherules. Note restricted width compared to much greater variability in length, consistent with string-like structure. Box shows quartiles, mean (circle), median (bold line), SD (whisker), min/max (x) for width and length. Outliers in red are from rod spherules with a single large gap junction. (**E**) Total gap junction length per rod for 42 rod spherules. Box shows quartiles, mean (circle), median (bold line), SD (whisker), min/max (x). (**F**) Average gap junction length plotted as a function of the number of gap junctions per rod shows that if there is only one gap junction per rod spherule, it is a long one. The two longest gap junctions from (**D**) were both singles (red, same two as in **D**). These are easily observed when labeled for Cx36 using confocal microscopy (*Figure 2—figure supplement 3*; *Jin et al., 2020*). The dashed line is the median gap junction length (0.419 µm), and it runs through the data for multiple gap junctions per rod. In other words, each additional gap junction is approximately a standard size. (**G**) Total gap junction length per rod spherule as a function of the number of gap junctions per rod shows that the single gap junctions (red) are outliers with great length. Total gap junction length tends to rise with the number of gap junctions. The straight line of linear regression shows the effect of adding a standard mean gap junction length (0.44 µm) each time and runs through all the data except for the singles (red). Fitted line is y = 0.44x; (R² = 0.33).

The online version of this article includes the following source data for figure 8:

**Source data 1.** Quantitative analysis of rod spherules from focused ion beam-scanning electron microscopy (FIB-SEM).

found there is usually clear separation between their membranes. Occasional small contacts between adjacent rod spherules show no membrane density, in contrast to rod/cone gap junctions (*Figure 9C*), and are often distant from the synaptic opening at the base where most Cx36 is clustered in the confocal data. Thus, our FIB-SEM data does not support the presence of rod/rod coupling in the mouse retina. This is consistent with physiological results that show the lack of direct rod/rod coupling (*Jin et al., 2020*).

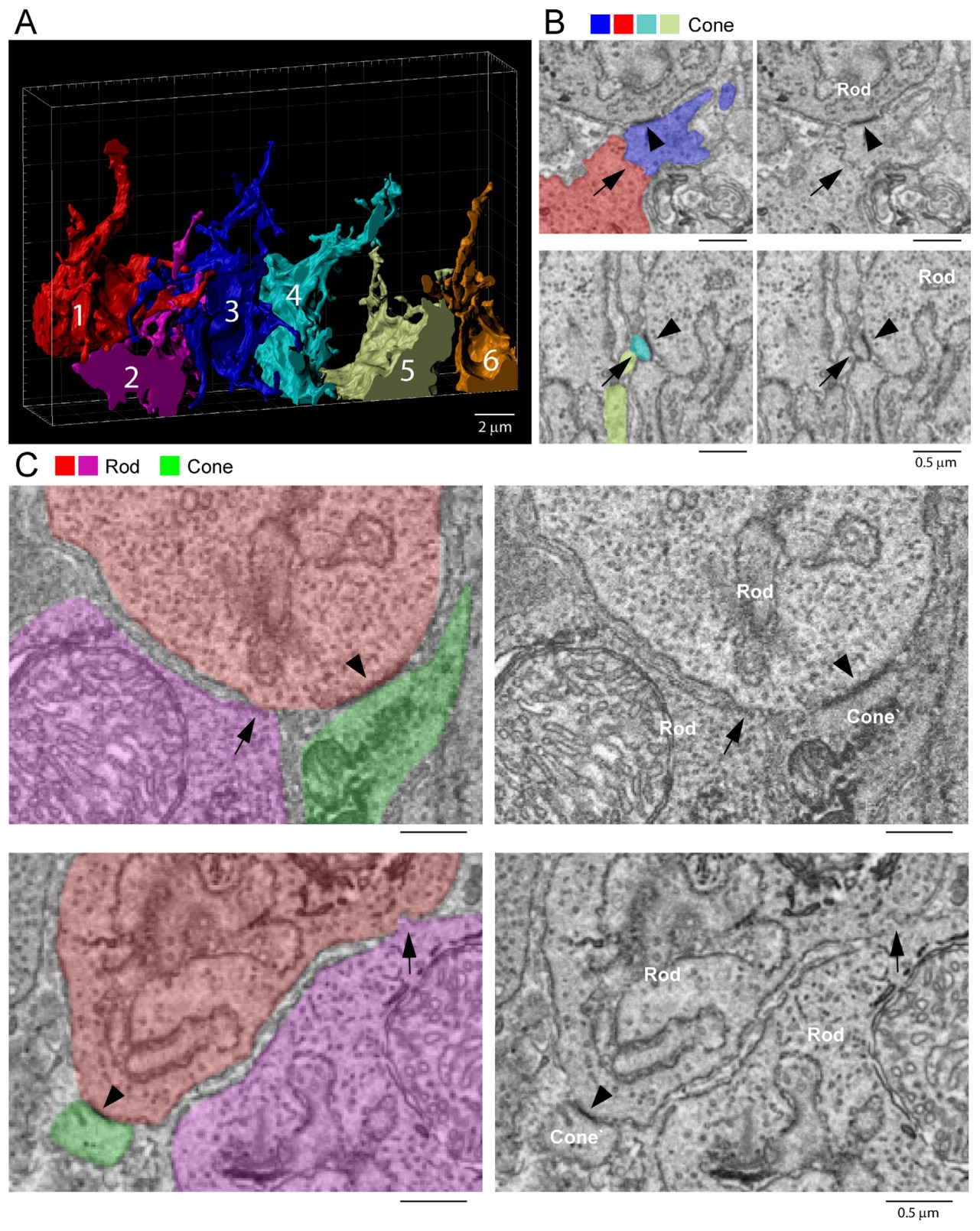

**Figure 9.** Rod/rod and cone/cone contacts do not appear as gap junctions, focused ion beam-scanning electron microscopy (FIB-SEM). (**A**) Partial reconstruction of six cone pedicles from FIB-SEM. (**B**) Examples of cone/cone contacts (red/blue or yellow/cyan, arrows), which show no membrane density. These examples were selected because of the nearby rod/cone contacts (arrowheads), which show prominent dense staining at their contact points, consistent with the presence of a gap junction. (**C**) Rod/rod contacts (red/magenta, arrows) show no membrane specialization compared to nearby rod/cone gap junctions (red/green, arrowheads).

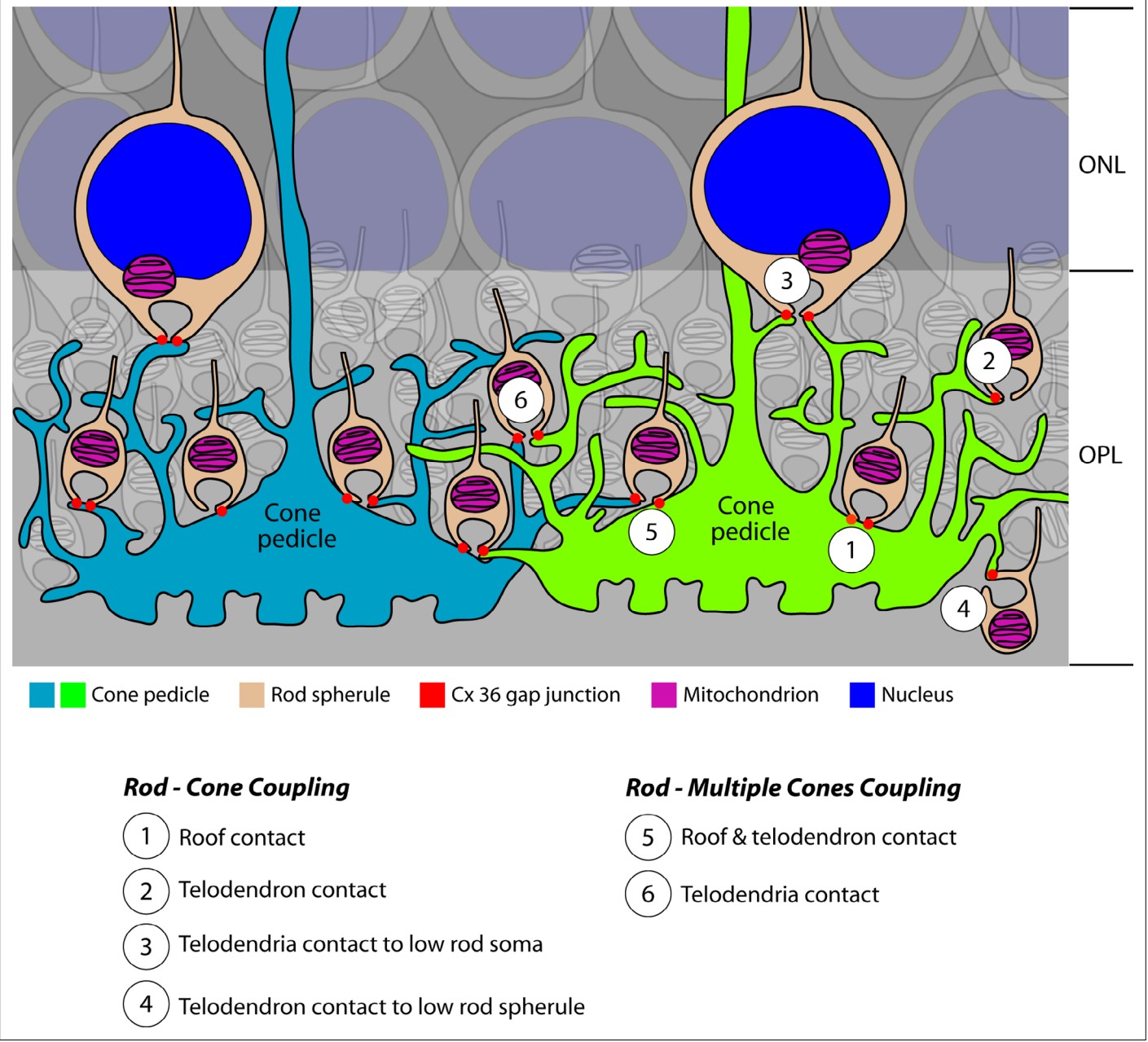

**Figure 10.** Cartoon summary of rod/cone gap junctions in the mouse retina. A cartoon showing the variety of Cx36 rod/cone gap junctions, including cone telodendrial contacts, cone pedicle roof contacts, and inverted rod spherules. The lowest row of rod cell bodies in the outer nuclear layer (ONL) do not have axons or rod spherules but still have Cx36 gap junctions with cone telodendria, which reach the upper margin of the outer plexiform layer (OPL). All these structural variants were also found with blue cone pedicles, indicating that there was no color specificity in rod/cone coupling. Most rod spherules have gap junction contacts with more than one cone. All rod spherules, with very few exceptions (such as areas of low cone density), make gap junctions with nearby cone pedicles. In contrast to the numerous rod/cone gap junctions, we could not detect rod/rod or cone/cone gap junctions in these experiments.

## Discussion

In this study, we demonstrate an approach to image and analyze small gap junctions reliably in a retinal neural circuit using a combination of light and electron microscopy, as summarized in *Figure 10*. For rod spherules, these sites of contact with cone telodendria correlate with the location of Cx36 immunofluorescence detected by confocal microscopy, both occurring close to the mouth of the synaptic opening, so there is a high probability that these sites represent gap junctions. By combining the

EM and immunofluorescence data with previously obtained electrophysiological measurements (*Jin et al., 2020*), we estimated the number of connexon channels and their conductance in the OPL of the mouse retina. Most importantly, our calculations suggest that the open probability for channels in a rod/cone gap junction may reach 100% with a dynamic range of ~20. Detailed calculations for the estimation are described below. By learning their basic properties, we have laid the groundwork to understand the role of rod/cone gap junctions in visual processing.

Because of their small size, gap junctions are often difficult to identify in EM micrographs, unless fortuitously captured in cross-section. Ideally, there should be sufficient resolution to reveal their pentalaminar signature, approximately 10 nm across (*Marc et al., 2018*; *Marc et al., 1988*). Indeed, some authors advocate resampling at high magnification (0.27 nm) with goniometric tilt to optimize visibility (*Sigulinsky et al., 2020*). However, the time investment makes this impractical to do for every potential gap junction in a sample. Thus, in spite of their importance, they are often left out of morphological analyses and physiological models due to the difficulty in assessing their numbers and dimensions from large-scale EM datasets (*Kasthuri et al., 2015*; *Scheffer et al., 2020*). In our FIB-SEM material, and appropriately fixed SBF-SEM material, the pentalaminar structure of gap junctions cannot be resolved, but an area of increased membrane density and chromophilic staining is present where the membranes of a cone telodendron and a rod spherule merge. The identity of the chromophilic material is unknown, but presumably it represents the aggregation of connexons and auxiliary proteins at gap junction sites (*Lasseigne et al., 2021*; *Nagy et al., 2018*). Correlation of the membrane densities in the FIB-SEM reconstructions with the confocal localization of Cx36 clusters was key to confirming the EM structures as gap junctions.

A similar approach could be applied to other systems to gain an understanding of gap junctions as circuit elements in specific neuronal pathways. The difficulty in finding and imaging gap junctions will largely be determined by their size, distribution, and abundance in a particular region (e.g., scattered distribution over a dendritic tree compared to precise localization on a small structure such as a rod spherule) and the morphological organization of the tissue (e.g., the stereotypical, layered organization of the OPL in retina versus a more complex morphology). Though it will be more challenging, we plan to apply our approach to the inner retina, where the circuitry is relatively complex and there is greater variability in gap junction morphology, from sparse strings of single connexons to crystalline plaques of much greater density (*Kamasawa et al., 2006*). More generally, our study demonstrates the utility of targeted, small-scale 'connectomic' analysis for the identification of neural circuit components. The promise of connectomics, to decode neuronal circuits and function, cannot be fulfilled without the inclusion of gap junctions and electrical coupling (*Scheffer and Meinertzhagen, 2021*).

## Rod/cone gap junctions predominate

Our results confirm the original work from cat retina, which showed 48 rod contacts per cone with a divergence of one rod to two cones (*Smith et al., 1986*), very close to the numbers reported here. In the primate retina, freeze-fracture EM showed string-like gap junctions of 400–600 nm in concentric arcs around the rod spherule synaptic opening (*Raviola and Gilula, 1973*). The telodendrial contacts described here have the same curved appearance and coincide with the location of Cx36 on the lower or vitreal surface of each rod spherule. With FIB-SEM, we found string-like rod/cone gap junctions with a mean length of ~480 nm, a tribute to the remarkable freeze-fracture EM work from nearly 50 years ago (*Raviola and Gilula, 1973*). Since that time, Cx36 has been identified as the major neuronal connexin and we show that Cx36 labeling signposts rod/cone gap junctions at the confocal level. In summary, the results from mouse, cat, primate, and human retinas all indicate that rod/cone gap junctions are a common feature of the mammalian retina (*Kántor et al., 2016*; *O'Brien et al., 2012*; *Raviola and Gilula, 1973*; *Smith et al., 1986*). We demonstrate that rod/cone coupling accounts for nearly all gap junctions between photoreceptors.

The position of rod/cone gap junctions, at the base of the rod spherule, close to the opening of the synaptic cavity, appears to be systematic in that the vast majority of rod/cone gap junctions occur at this site. Our results show a range of 1–6 Cx36 contacts per rod spherule, comparable with 4–6 cone contacts per rod in the cat retina, forming a ring of contacts around the postsynaptic opening of the rod spherule (*Smith et al., 1986*). In rod spherules from the mouse retina, two gap junction contacts were described, usually on opposite sides of the synaptic opening, mostly arising from a single cone (*Tsukamoto et al., 2001*). We may speculate that gap junctions are localized with some of the same

scaffolding proteins that occur at the rod synaptic terminal, but the functional significance of this repeated motif is unknown. In mutant mouse lines, where Cx36 has been deleted from either rods or cones, cone telodendria are still present and they still reach out to contact nearby rod spherules in the absence of rod/cone gap junctions (*Jin et al., 2020*). Therefore, the specificity of synaptic connections is not determined or maintained by the presence of Cx36 gap junctions.

Previously, some Cx36-GFP labeling was also found in the ONL (*Feigenspan et al., 2004*), suggesting that some coupling may occur between somas and/or passing axons. However, we did not observe this pattern with Cx36 antibodies and, in our hands, Cx36 labeling of photoreceptors was restricted to the OPL (*Figure 1*). It is possible that the Cx36-GFP construct caused a trafficking defect.

Rod/cone coupling forms the entry to the secondary rod pathway (*Kolb, 1977*; *Smith et al., 1986*; *Bloomfield and Dacheux, 2001*). It was previously thought that these gap junctions must be closed at night to preserve the amplitude of single-photon responses of individual rods, which underlie scotopic sensitivity (*Smith et al., 1986*). However, recent work indicates that rod/cone coupling is high at night and low in daytime due to circadian activity and the influence of dopamine (*Jin and Ribelayga, 2016*; *Jin et al., 2020*). This leads to a reduction in the amplitude of the single-photon response at night. While this may appear to be detrimental, we have suggested that noise reduction, a consequence of convergence and coupling in the photoreceptor network, is the major function of rod/cone gap junctions, in addition to driving the secondary rod pathway (*Field et al., 2019*; *Jin et al., 2020*).

## Rods and cones both express Cx36

The small number of Cx36 clusters per rod (2–3), compared to more than 50 per cone pedicle, a ratio of approximately 20, may explain early failures to detect rod Cx36 RNA transcripts (*Bolte et al., 2016*; *Feigenspan et al., 2004*). However, there is strong physiological evidence that Cx36 is required for rod/cone coupling (*Asteriti et al., 2017*; *Ingram et al., 2019*; *Jin et al., 2020*). Cx36 is the most common neuronal connexin, and it does not pair with other connexins, making only homotypic gap junctions (*Koval et al., 2014*; *Nagy et al., 2018*; *Teubner et al., 2000*). We have recently shown that both rods and cones express Cx36 and that no other connexins are present in either rods or cones (*Jin et al., 2020*). In addition, Cx36 is required on both sides to form a rod/cone gap junction (*Jin et al., 2020*; *Miller et al., 2017*). Thus, the simplest case holds that both rods and cones express Cx36, and rod/cone gap junctions are heterologous but homotypic (meaning between different cell types but both expressing the same connexin).

## Cx36 numbers for rods and cones are consistent

We gathered data for both rods and cones independently. Because they are linked by rod/cone gap junctions, they should each serve as a check on the other. From confocal imaging of individual EGFP-labeled cone pedicles, we estimate that each cone pedicle has 51.4 ± 8.88 (mean ± SD, n = 18) Cx36 clusters (*Figure 2*), presumed to represent gap junctions, with nearby rod spherules. From the rod perspective, 43 rods contact each cone, and there are 2.48 ± 1.01 (mean ± SD, n = 260 rods, *Figure 3F and G*) Cx36 clusters at the base of each rod spherule. Most rods contact more than one cone, with a mean divergence of 1.89 (*Figure 5E*, *Appendix 2—table 1*); thus, the number of rod/cone gap junctions per cone pedicle can be calculated as 43 × 2.48/1.89 = 56.4, in close agreement with the number of Cx36 gap junctions counted in confocal reconstructions of individual cone pedicles. These numbers are also close to previous calculations of ~45 Cx36 clusters/cone pedicle, based on the density of Cx36 clusters and the number of cone pedicles from wholemount retina (*Jin et al., 2020*).

The number of gap junctions per rod was calculated by two methods: a confocal analysis of Cx36 clusters and using the FIB-SEM data. Analysis of the FIB-SEM data gave 3.21 gap junctions per rod spherule. The confocal analysis of Cx36 clusters gave 2.48 gap junctions per rod spherule, 25% less. The lower number in the confocal analysis might be expected because small Cx36 clusters could be missed, and two close together could merge and be counted as one due to the lower resolution of light microscopy (*Sigulinsky et al., 2020*). Despite the differences in resolution, both methods showed a similar number of gap junction contacts close to the rod postsynaptic compartment. Although the smallest Cx36 clusters may be less than the confocal detection limit (different from the resolution limit), these data suggest that we accounted for at least 75% of the photoreceptor gap junctions by confocal microscopy.

## Blue cone pedicles are also coupled to rods

In the coupled cone networks of primate and ground squirrel retina, there is good evidence that blue cones are not coupled to neighboring red/green (primate) or green cones (ground squirrel) (*Hornstein et al., 2004*; *Li and DeVries, 2004*; *O'Brien et al., 2012*). In the primate retina, the telodendria of blue cones are few in number and too short to reach the neighboring red/green cones (*O'Brien et al., 2012*). Thus, blue cones appear to be electrically separated from other cones in these two species, perhaps to maintain spectral discrimination (*Hsu et al., 2000*). In the mouse retina, although the blue cones were identified by *Behrens et al., 2016*, we were unable to find any cone-to-cone gap junctions, regardless of spectral sensitivity (see below).

In contrast to the selective connections between cones in some species, rods were coupled to both blue and green cones indiscriminately in the mouse retina (present work) and in primate retina (*O'Brien et al., 2012*). Blue cones, identified in confocal work by the presence of S-cone opsin, and in SBF-SEM by their connections with blue cone bipolar cells (*Behrens et al., 2016*; *Nadal-Nicolás et al., 2020*), and green cones both made telodendrial contacts at Cx36 clusters with all nearby rod spherules (*Figure 5*). Thus, we find no evidence for color specificity in rod/cone coupling. In fact, a single rod spherule may be coupled to both blue and green cones (*Figure 6—figure supplement 5*). Therefore, rod signals can pass via the secondary rod pathway into both blue and green cones and their downstream pathways. Considering blue cone circuits specifically, rod input to blue cone bipolar cells and downstream circuits is predicted via the secondary rod pathway, in addition to the previously reported primary rod pathway inputs from AII amacrine cells to blue cone bipolar cells (*Field et al., 2009*; *Whitaker et al., 2021*).

## Cone/cone gap junctions are rare

We were unable to confirm the presence of cone/cone gap junctions in the mouse retina despite locating more than 20 examples of cone/cone contacts from six partially reconstructed cone pedicles from one FIB-SEM dataset. In each case, there was no indication of the typical membrane density or chromophilic staining at cone/cone contacts. Nearby rod/cone contacts provided prominent gap junctions for comparison (*Figure 9B*). This result is surprising, given previous reports of cone/cone coupling in several mammalian species, including mouse (*DeVries et al., 2002*; *Kolb, 1977*; *Smith et al., 1986*; *Tsukamoto et al., 2001*). However, close contacts without gap junctions are not unprecedented. In rabbit retina, many bipolar cells are coupled by gap junctions, but some are not, despite contacts and the presence of gap junctions elsewhere in the same cell (*Sigulinsky et al., 2020*). Gap junctions do not determine the specificity of neuronal connections.

The available data was derived from a limited set of six partially reconstructed cones due to the high-resolution, yet small volume of the FIB-SEM dataset. Thus, while we are confident that rod/cone coupling accounts for the vast majority of photoreceptor gap junctions, the sample size is too small to rule out a minor amount of cone/cone coupling, perhaps too small or too faintly stained to be readily detected by confocal imaging. Our previous electrophysiological studies suggest that there is weak cone-to-cone coupling in the mouse retina, which may indicate that cone/cone gap junctions are smaller than rod/cone gap junctions. Cone/cone coupling persists in the rod-specific Cx36 KO, suggesting it is direct (*Jin et al., 2020*). In the rod-specific Cx36 KO, there was some residual Cx36 labeling, associated with cone telodendria, which was significantly different from the background noise and may represent a small amount of cone/cone coupling, estimated as two gap junctions/cone (*Jin et al., 2020*).

Cone-to-cone coupling has been reported in other mammals but there may be species variation. In the ground squirrel, a cone-dominated retina, the low number of rods with the resulting adjacency of the cones may promote cone/cone coupling (*DeVries et al., 2002*; *Li, 2020*; *Li et al., 2010*). In the central primate retina, cones are densely packed and often adjacent, which is not the case in mouse retina. In peripheral primate retina, cones are more widely spaced, and they are connected by a sparse array of telodendria, which seem to target neighboring cones, making the pattern of Cx36-labeled gap junctions very obvious, in addition to numerous rod/cone gap junctions (*O'Brien et al., 2012*). Finally, in the rod-less, cone-only mouse, there is a large increase in Cx36 labeling in the OPL, suggesting that cone/cone coupling occurs, at least under these circumstances (*Dang et al., 2004*). The evidence for photoreceptor coupling in our study, including weak cone/cone coupling, is summarized in Appendix 1.

## No rod/rod gap junctions

Previous work in the mouse retina has suggested that rods form a coupled network based on the appearance in electron micrographs of small contacts between rods (*Tsukamoto et al., 2001*). Most of these rod/rod contacts were characterized as small and with no membrane density (*Tsukamoto et al., 2001*). Furthermore, some of the rod/rod contact sites were between the spherules and passing rod axons, a location where there is no Cx36 labeling. We have searched diligently for rod/rod gap junctions, but we have been unable to confirm their presence. Not only is Cx36 labeling restricted to the base of the rod spherule, around the synaptic opening, but, in the FIB-SEM material, this was almost the only location where we found rod gap junctions. We mapped the gap junctions from 42 rod spherules, and their distribution was in close agreement with the distribution of Cx36 labeling and cone telodendrial contacts. We could not trace every process making a gap junction contact to their cell of origin, but in 112 cases where we could, all were cone telodendria. Even the small number of outlying gap junctions distant from the synaptic opening were traced to cones.

We did find small contacts between adjacent rod spherules (*Figure 9C*), but there was no membrane density indicating the presence of a gap junction, and, in this equatorial position, around the midline where the rod spherules are closely packed, there was no Cx36 labeling. Furthermore, apparent rod/rod coupling was eliminated in the cone-specific Cx36 KO, indicating that physiological evidence of rod/rod coupling in wildtype retina is actually due to indirect rod/cone/rod coupling via the network (*Jin et al., 2020*). In the cone-specific Cx36 KO, which should preserve rod/rod gap junctions, Cx36 labeling of photoreceptors was essentially eliminated, providing no evidence for residual rod/rod coupling (*Jin et al., 2020*). The loss of most Cx36 coupling in either the rod- or cone-specific KO indicates that rod/cone coupling accounts for the majority of photoreceptor gap junctions and provides no support for direct rod/rod coupling. We are aware of the difficulty in proving the absence of a particular structure, but, based on the weight of the available evidence, we conclude that there are no rod/rod gap junctions in the mouse retina, and that this may be a common feature of the mammalian retina. The comparative evidence for photoreceptor coupling, including the lack of rod/rod coupling, is summarized in Appendix 1.

## Calculations of size for rod/cone gap junctions

Gap junctions are common building blocks of neural circuits throughout the CNS, but to accurately model their effect on circuit function, it is essential to know their size, number of connexon channels, their conductance, and perhaps most importantly, their dynamic range or plasticity. The size of the conductance through a gap junction depends not only on the number of connexons and the unitary conductance, but also on the channel activity, usually characterized as the open probability. These are the fundamental properties required to understand the function of gap junctions in neuronal microcircuits. Here, we have an opportunity to estimate these variables.

Rod/cone gap junctions appear as low-density strings of single particles in freeze-fracture EM (*Raviola and Gilula, 1973*; *Reale et al., 1978*), and this is consistent with the uniform width of rod/cone gap junctions in our 3D FIB-SEM reconstructions, as well as the low intensity of Cx36 labeling compared to the much larger plaque-type gap junctions of the inner retina (*Kamasawa et al., 2006*). Therefore, in our calculations, we have taken rod/cone gap junctions as strings with a width of one connexon and a length given by the 3D reconstructions of gap junctions from the FIB-SEM datasets.

To calculate the number of connexons from the length of gap junctions, we need to determine the spacing of Cx36 channels. The mean density of gap junction plaques in freeze-fracture preparations of mouse cerebellum was $12,940 \pm 405/\mu m^2$ (mean ± SEM, n = 12), close to the density of Cx36 plaques in the inner retina ($12,000/\mu m^2$) (*Kamasawa et al., 2006*; *Szoboszlay et al., 2016*). From this, we calculated the center-to-center spacing as $9.45 \pm 0.296$ nm (mean ± SEM) for connexons in a Cx36 gap junction plaque. This result is in close agreement with 10 nm spacing reported in freeze-fracture material of rod/cone gap junctions in primate retina (*Raviola and Gilula, 1973*).

Knowing the spacing, we calculated the number of connexon channels from the mean length of a rod/cone gap junction in the FIB-SEM dataset, $477 \pm 19.5$ nm (mean ± SEM, n = 135 from 42 rod spherules), yielding $50.5 \pm 2.60$ (mean ± SEM) connexon channels per rod/cone gap junction. We estimate that there are $3.21 \pm 0.190$ (mean ± SEM, n = 42, *Appendix 2—table 3*) gap junctions per rod, but these are shared with several nearby cones. Dividing by the rod to cone divergence (the number of cones contacted by each rod), $1.89 \pm 0.0337$ (mean ± SEM, n = 361, *Appendix 2—table 3*), gives

1.70 ± 0.105 (mean ± SEM) gap junctions or a mean of 85.9 ± 6.91 (mean ± SEM) connexons between an average rod/cone pair. For a unitary conductance of 14.3 ± 0.8 pS (mean ± SEM, n = 92, per Cx36 channel) (*Srinivas et al., 1999*; *Teubner et al., 2000*), we can calculate the maximal conductance between a rod and a cone, if all gap junction channels were in an open state, as ~1228 ± 120 pS (mean ± SEM, 85.9 connexons × 14.3 pS/connexon). It is important to note that this theoretical maximum was derived from our morphological data.

## Possible sources of error

It is important to note that there are several assumptions and potential sources of error that may affect these calculations. The structure of small gap junctions lies below the resolution of conventional light microscopy and must be inferred. Likewise, in EM material, the identity of the darkly stained chromophilic material is unknown and may include auxiliary proteins in addition to connexins, which may exaggerate the dimensions of a gap junction, particularly in width for string-like gap junctions. For this reason, our estimates of rod/cone gap junction width were guided by freeze-fracture SEM images from primate (*Raviola and Gilula, 1973*; *Reale et al., 1978*). Thus, our calculations are based on the length of gap junction strings with a width of one connexon because the freeze-fracture studies show a string of single-gap junction particles on the surface of a rod spherule (*Raviola and Gilula, 1973*; *Reale et al., 1978*). This is consistent with the low intensity of Cx36 labeling in the OPL, which indicates the presence of very small or sparse gap junctions such as strings (*Dang et al., 2004*; *Mills et al., 2001*). If we failed to detect some gap junctions, this would artificially increase the open probability numbers. However, this is probably a minor concern because we were able to detect gap junctions with a minimum length of approximately 100 nm, or ~10 connexons, in the FIB-SEM material (*Figure 8*). These issues of gap junction size and string width could perhaps be addressed in the future by super-resolution microscopy for Cx36.

We are assuming negligible species variation in the size and packing density of rod/cone gap junction channels, and this is supported by the similarity in measurements of rod/cone gap junction length, ~500 nm, from our FIB-SEM data in mouse and the freeze-fracture studies in primate (*Raviola and Gilula, 1973*; *Reale et al., 1978*). Tissue shrinkage during EM sample processing is another potential source of error, though our measurements of gap junction position from SEM and confocal microscopy gave comparable results (*Figures 3E, 8B and C*). Therefore, the errors introduced by the use of these different microscopy techniques are unlikely to be large and do not materially affect our conclusions.

Finally, previous work reported that electrical coupling between photoreceptors changes with the time of day, reflecting the influence of light/dark adaptation and circadian clocks (*Ribelayga et al., 2008*; *Li et al., 2013*; *Zhang et al., 2015*; *Jin et al., 2015*; *Jin and Ribelayga, 2016*; *Jin et al., 2020*). Yet, daily changes in photoreceptor coupling reflect changes in Cx36 phosphorylation state and open probability, without significant structural changes in the number or size of Cx36 gap junctions present (*Li et al., 2013*; *Zhang et al., 2015*). Thus, circadian changes are not likely to affect our conclusions concerning the structure of rod/cone gap junctions.

While we have listed these potential sources of error in the interest of transparency, we note that a combined underestimation of the connexon number by a factor as large as two yields a maximum open probability number of >50% (107%/2). This would still provide a historically high number for the open probability of gap junctions. Thus, we believe our calculations predict that the fraction of connexons that participate in gap junction modulation is greater than previously appreciated. Our results suggest that modulation of gap junction strings may include most connexon channels, perhaps approaching 100% when pharmacologically driven to an extreme.

## Dynamic range of rod/cone gap junctions

Using this detailed structural information, combined with our previous conductance measurements from rod/cone paired recordings, we can calculate the physiological properties of rod/cone gap junctions. Our recently published work shows a mean resting value of 307 ± 2.31 pS (mean ± SEM) for the rod/cone transjunctional conductance (*Jin et al., 2020*). Compared to the theoretical maximum above, this indicates a resting open probability in darkness of 25.0% ± 2.45% (mean ± SEM). Notably, this is much higher than previous estimates of around 1% (*Connors, 2017*; *Marandykina et al., 2013*). But these earlier values may be low because the size of the gap junctions was estimated by

immunofluorescence, which provides a high number for gap junction area and connexon number, with a correspondingly low number for the open probability (*Kamasawa et al., 2006*; *Szoboszlay et al., 2016*). More recently, when the number of connexons was accurately measured by freeze-fracture EM, the open probability of cerebellar gap junctions was calculated as 18% (*Szoboszlay et al., 2016*), close to the value derived here.

Rod/cone coupling is modulated by dopamine (*Ribelayga et al., 2008*; *Jin et al., 2020*). From our previous studies with paired recordings, in the presence of quinpirole, a D2 agonist, rod/cone coupling was measured as 35.5 ± 0.968 pS (n = 500, bootstrapping from data in *Jin et al., 2020*). In the presence of spiperone, a D2 antagonist, the maximum conductance for a rod/cone pair was 1312 ± 22.8 pS (n = 500, bootstrapping from data in *Jin et al., 2020*). Compared to the theoretical maximum of 1228 ± 120 pS above, derived from morphological data, these values translate to a minimum open probability of 2.89% ± 0.293% and a maximum open probability of 107% ± 10.6% of available Cx36 gap junction channels. Note that details of these calculations with the cumulative errors can be found in Appendix 3. This is a very surprising result because previous measures of open probability have produced such low estimates (*Connors, 2017*), but we believe it is consistent with our data. For the first time, it suggests that the range of gap junctions can be modulated from close to 0 to approximately 100% of available Cx36 channels. In other words, all of the channels in these small string-like rod/cone gap junctions are switchable and may contribute to their plasticity.

The diffusion coefficient through Cx36 gap junctions, a proxy for conductance, was correlated with the phosphorylation of Cx36 as measured using phospho-Cx36 antibodies (*Kothmann et al., 2009*; *O'Brien, 2014*). Pharmacological manipulation using dopamine agonists or antagonists produced phosphorylation-driven changes in tracer coupling between mouse photoreceptors that encompassed a 20-fold range of diffusion coefficients, producing a large dynamic range for gap junction plasticity (*Li et al., 2013*). In the present experiments, the ratio of minimum to maximum open channels, from 3–100% or 36–1312 pS, was approximately 30, similar to the dynamic range derived from tracer coupling studies with phospho-specific Cx36 antibodies. While these calculations are necessarily approximate, they support the concept that gap junctions provide a versatile component, offering a large range of plasticity in neural circuits. In the retina, rod/cone coupling may reduce transduction noise in the photoreceptor network and the modulation of rod/cone gap junctions also provides a switchable entry to the secondary rod pathway, which varies with light intensity and the circadian cycle (*Bloomfield and Völgyi, 2009*; *Field et al., 2019*; *Jin et al., 2020*; *Jin and Ribelayga, 2016*). In more general terms, this example from the retina demonstrates that gap junctions may perform a variety of essential functions in neural circuits.

## Materials and methods
### Animals
All animal procedures were reviewed and approved by the Animal Welfare Committee at the University of Texas Health Science Center at Houston (AWC-20-0138), Oregon Health & Science University (IP00000456, A3304-01), and University of Maryland College Park (R-OCT-20-56). C57BL/6J (stock no. 000664) mice were purchased from the Jackson laboratories. We used mice 2–6 months of age of

**Table 1.** Antibodies.

| Antibody | Source | Catalog # | Species | Dilution | Notes |
|---|---|---|---|---|---|
| Cone arrestin | Millipore | AB15282 | Rabbit | 1:1000 | Labels cones, including pedicle |
| Cx36 | Millipore | MAB3045 | Mouse | 1:1000 | Cx36 gap junctions |
| vGlut1 | Synaptic Systems | 135304 | Guinea pig | 1:3000 | Labels PR terminals, especially rod spherules |
| Blue cone opsin | Millipore | AB5407 | Rabbit | 1:1000 | Blue cone outer segments |
| Ribeye/CtBP2 | BD Transduction | 612044 | Mouse | 1:500 | Synaptic ribbon marker |
| TOMM20 | Santa Cruz | SC-11415 | Rabbit | 1:1000 | Mitochondrial marker |
| mGluR6 | Generous gift from Kiril Martemyanov | | Sheep | 1:1000 | mGluR6 receptors of ON bipolar cells and rod bipolar cells |
| PMCA | Santa Cruz Biotechnology | sc-28765 | Rabbit | 1:200 | Labels rod spherule plasma membrane |

either sex. Animals were housed under standard laboratory conditions, including a 12 hr light/12 hr dark cycle. All animals were euthanized in the middle of the day (during the light cycle), under room lights.

## Antibodies and immunocytochemistry

Mice were anesthetized by intraperitoneal injection of a ketamine/xylazine mix solution (100/10 mg/kg) before being euthanized by cervical dislocation. Eyes were rapidly collected, hemisected, the vitreous was removed, and the resulting eyecup was placed in 4% paraformaldehyde in phosphate-buffered saline (PBS) at room temperature for 1–2 hr. Vibratome sections (Leica VT 1000S) or wholemounted retinas were reacted with a cocktail of antibodies (*Table 1*), according to procedures described previously (*Li et al., 2013*; *O'Brien et al., 2012*; *Jin et al., 2020*). Briefly, sections were washed and blocked in 3% donkey serum/0.3% Triton X-100 (in PBS) for 2 hr (overnight for whole mounts) and incubated overnight at room temperature with a cocktail of primary antibody(ies) in 1% donkey serum/0.3% Triton X-100 (in PBS) (7 days for whole mounts). Tissues were processed free-floating on an oscillating platform at 1 Hz. Following incubation with the primary antibody, sections were rinsed in PBS (6×, 20 min) and reacted with a secondary antibody(ies) for 2 hr (overnight for whole mounts) at room temperature in the dark. Donkey Alexa Fluor–, Cy3-, or DyLight-conjugated secondary antibodies were purchased from Jackson ImmunoResearch Laboratories Inc (West Grove, PA) and used at 1:600 dilution. Last, sections or whole mounts were covered with mounting medium and sealed with nail polish. 4′,6-diamidino-2-phenylindole (DAPI) (100 µg/ml) was added to the mounting medium to stain the nuclei.

## Blue cone opsin Venus mouse line

A bacteria artificial chromosome (BAC) clone (bMQ-440P15) from an SV129 genomic library (bMQ), encompassing the entire mouse Opn1sw locus, was used to create the Opn1sw_Venus BAC transgenic construct through ET recombination as described elsewhere (PMID: 9771703). Briefly, a Venus-bGH PolyA-NeoR (flanked by FRT sites) cassette was integrated into the Opn1sw locus on the BAC clone replacing coding region of exon 1 to generate the Venus-bGH-Neo knockin version of the BAC clone. Recombination was verified by PCR reactions specifically designed to detect recombination junctions by ET recombination (or homologous recombination). This BAC clone carrying the reporter cassette was further trimmed, through ET recombination, to retain a 15 kb fragment, including upstream regulatory sequence (10 kb upstream of the coding region), the KI cassette, and the ~2 kb sequence downstream of exon 2 of the Opn1sw gene. The intended transgenic construct was verified by overlapping PCR and Sanger sequencing. Transgenic F0 founders were created by microinjecting the transgenic construct (1 ng/µl) into fertilized eggs from C57bl6/j donors followed by genomic PCR screening amplifying the reporter insert.

## Confocal microscopy image acquisition

A Zeiss LSM-800 confocal microscope with Airyscan was used for four-channel imaging of retinal whole mounts and sections with a ×63 (N.A. 1.4) oil-immersion objectives. Images with 30–40 nm pixel size were acquired in series of 0.15 µm optical sections. Airyscan images were processed by Zen software (Zeiss) to yield super-resolution images. We measured the full width at half maximum of the point spread function for this instrument as 170 nm using 0.1 µm latex beads (Molecular Probes). Figures were presented as short stacks of 2–6 images. Images were processed in ImageJ, Imaris (Oxford Instruments), or Photoshop (Adobe Systems Inc) for contrast enhancement and further analysis.

## Quantitative analysis of Cx36 clusters

### Single cones

Retinal sections from OPN4-EGFP mice were used because only a few cones were EGFP positive and it was possible to image well-isolated single-cone pedicles. Airyscan images were loaded in Imaris software, and the colocalization between the cone (EGFP) and Cx36 (Cy3) was analyzed. Neighboring colocalized voxels were grouped as a 3D object by surface rendering, and the number of extracted 3D objects was measured.

### Rod spherules

An antibody against the synaptic vesicular glutamate transporter (vGlut1) was used to label rod spherules. It does not stain the cell membrane and thus underfills rod spherules (*Quraishi et al.,*

*2007*). Some sections were also stained for Plasma Membrane Calcium ATPase (PMCA), which stains the rod spherule membrane (*Johnson et al., 2007*), encircling the vGlut1 labeling (*Figure 6—figure supplement 3A*). Due to lack of clear colocalization between Cx36 and the rod marker (vGlut1), the number of Cx36 clusters per rod was counted manually. Some rod spherules were excluded from analysis if they were too close to link a Cx36 cluster to a specific rod. A total of 260 well-isolated rod spherules in seven retinal sections from three retinae were chosen to analyze the number of Cx36 clusters. To analyze the position, a mini-stack of 6 × 0.15 µm optical sections from 18 rod spherules wase extracted, aligned, and averaged (ImageJ). A spline curve 8 pixels (272 nm) wide was applied to the perimeter, and the intensity profile for Cx36 was plotted linearly around the synaptic opening.

## Connectivity analysis: SBF-SEM (e2006)

A region in the OPL from a publicly available serial block-face scanning EM (SBF-SEM) dataset e2006, voxel size 16.5 × 16.5 × 25 nm (*Behrens et al., 2016*; *Helmstaedter et al., 2013*), including 164 cone pedicles, was analyzed using KNOSSOS software (*Helmstaedter et al., 2011*) (https://knossos.app/). A volume of approximately 15 µm (OPL depth) × 50 µm × 50 µm in the OPL was chosen to analyze, in which we identified 29 cone pedicles and 811 rod spherules. Because there are some areas of relatively low cone density, an additional area, including six widely spaced cone pedicles, was also analyzed. The whole e2006 dataset included a total of six blue cones (*Behrens et al., 2016*) but one was located at the edge of the dataset and was truncated. Thus, a total of five blue cones were analyzed.

### Skeletons

We traced the telodendria from all 29 cone pedicles by skeletonization with KNOSSOS. All rod spherules contacted by a specific cone were identified. We used the membrane apposition between cone pedicle/telodendria and rod spherule to indicate rod/cone contact. The contact information was stored as an annotation directory in KNOSSOS and further analyzed using Excel (Microsoft) and Origin (OriginLab Corp).

### Segmentation and 3D reconstruction

A volume of 27 µm (OPL plus a part of ONL) × 27 µm × 39 µm was extracted from the e2006 dataset to segment using Microscopy Image Browser (*Belevich et al., 2016*). The segmented voxel data were loaded into Imaris to create a 3D reconstruction by surface rendering. Images were constructed in Imaris by rotating and adjusting transparency as required.

## SBF-SEM: Singer dataset (eel001)

An excised retina was fixed for 1 hr at room temperature with 2% glutaraldehyde in 0.15 M caco-dylate buffer, washed in three changes of the same buffer, and postfixed with 1% osmium tetroxide in 0.15 M cacodylate containing 1.5% potassium ferrocyanide. A wash in three changes of distilled water followed the reduced osmium fixation and preceded an en bloc fix in 2% aqueous uranyl acetate. Dehydration in a graded series of ethanol (35–100%) and infiltration in a propylene oxide:epoxy resin series was followed by embedding and polymerization in epoxy resin. Imaging by SBF-SEM was performed under contract with Renovo Neural (Cleveland, OH) using a FEI Teneo Volume Scope. Beam current was 50 pA, and the face was imaged with 7 × 7 × 40 nm resolution. Each face image spanned the width of the retina from the first layer of photoreceptor cell bodies in the ONL to the ganglion cell bodies in the GCL and comprised two stitched fields of 8192 × 8192 pixels for a dimension of 8192 × 16,384 pixels or 57.34 × ~114.68 µm. In total, 1649 slices were acquired, making the z-depth traversed = 65.96 µm. Image stacks were viewed using KNOSSOS and processed in Photoshop.

## FIB-SEM sample preparation

Mice were euthanized and enucleated; retinas were quickly dissected and placed in 3% paraformaldehyde and 1% glutaraldehyde for 30 min, then processed for electron microscopy using the Dresden protocol (*Paridaen et al., 2013*). The resulting resin blocks were trimmed to contain the OPL within 25 µm of the top edge of the block, cut and mounted to a 45° pre-tilted stub, and coated with 8 nm of carbon. Three-dimensional data were acquired using the Helios G3 NanoLab DualBeam FIB-SEM. In brief, a focused beam of gallium atoms ablated 4 nm off the surface of the sample (FIB conditions:

30 keV accelerating voltage, 790 pA beam current), comprising the scanning electron microscope (SEM) imaging area, with a field of view around 25 μm. The freshly ablated surface was then imaged by backscattered electrons using the In-Column Detector (SEM conditions: 400 pA beam current, 3 keV accelerating voltage, 4 nm per pixel, and 3 μs dwell time; image resolution: 6144 × 4086, 4 nm isotropic). Around 1149 (FIB-SEM 1 dataset; 1342 for FIB-SEM 2 dataset) ablation and imaging cycles were run over a 3-day period, resulting in a sample depth of roughly 4.6 μm. Images were then registered using the Linear Stack Alignment with SIFT algorithm (Fiji) and cropped to remove artifacts arising from the alignment.

## Gap junction position and size analysis from FIB-SEM

Individual rod spherules were cropped from the datasets and segmented for 3D reconstruction as described above for the e2006 dataset. Gap junctions were identified by a darkly stained area of merged membranes. The distance from a gap junction to the synaptic opening center of a rod spherule was calculated using the oblique slicer tool in Imaris to align the center of the mitochondria and the postsynaptic opening in a single plane. Gap junction length was visualized using the oblique slicer to create an en face view. Some gap junctions showed belt-like structures with curvature and torsion. Due to the complexity of the structure, the size of these gap junctions was estimated as a simplified rectangular shape by measuring the length of the longer axis and average width of the belt. The acquired data were analyzed in Excel and Origin.

## Acknowledgements

This project was inspired by the paper from Behrens et al., 2016, who used e2006 to reconstruct bipolar cells. We thank Christian Behrens, Timm Schubert, Philipp Berens, and Thomas Euler (University of Tübingen) for generously sharing data on blue cone bipolar cells. We thank Moritz Helmstaedter (MPI, Frankfurt) for hosting the e2006 dataset. We thank Kiril Martemyanov (Scripps research Institute, Jupiter, Florida) for the generous gift of an mGluR6 antibody. We thank David Berson (Brown University) for advice, encouragement, and an introduction to connectomics. We thank Jessica Riesterer at the Multiscale Microscopy Core, an OHSU University Shared Resource core facility, for acquiring the FIB-SEM datasets. We thank Alice Chuang (Richard Ruiz Department of Ophthalmology and Visual Science, McGovern Medical School) for statistical analysis. This work was supported by NIH grants EY017836 (JHS); EY029408 (SCM and CPR); P30EY028102 (SCM); P30NS061800 (SAA); RF1MH127343 (SAA, SCM, CWM, and CPR).

## Additional information

### Funding

| Funder | Grant reference number | Author |
|---|---|---|
| National Institute of Mental Health | RF1MH127343 | Catherine W Morgans<br>Sue A Aicher<br>Christophe P Ribelayga<br>Stephen C Massey |
| National Eye Institute | EY029408 | Christophe P Ribelayga<br>Stephen C Massey |
| National Eye Institute | EY017836 | Joshua H Singer |
| National Institute of Neurological Disorders and Stroke | P30NS061800 | Sue A Aicher |
| National Eye Institute | P30EY028102 | Stephen C Massey |

The funders had no role in study design, data collection and interpretation, or the decision to submit the work for publication.

## Author contributions

Munenori Ishibashi, Conceptualization, Data curation, Formal analysis, Investigation, Methodology, Validation, Visualization, Writing – original draft, Writing – review and editing; Joyce Keung, Investigation, Methodology, Validation; Catherine W Morgans, Sue A Aicher, Conceptualization, Formal analysis, Funding acquisition, Investigation, Methodology, Project administration, Resources, Validation, Visualization, Writing – original draft, Writing – review and editing; James R Carroll, Formal analysis, Investigation, Methodology, Resources, Visualization; Joshua H Singer, Funding acquisition, Investigation, Methodology, Resources, Writing – review and editing; Li Jia, Resources; Wei Li, Resources, Writing – review and editing; Iris Fahrenfort, Conceptualization, Investigation, Methodology, Writing – original draft, Writing – review and editing; Christophe P Ribelayga, Conceptualization, Formal analysis, Funding acquisition, Investigation, Project administration, Supervision, Writing – original draft, Writing – review and editing; Stephen C Massey, Conceptualization, Formal analysis, Funding acquisition, Investigation, Methodology, Project administration, Supervision, Validation, Visualization, Writing – original draft, Writing – review and editing

## Author ORCIDs

Munenori Ishibashi http://orcid.org/0000-0002-6922-573X
James R Carroll http://orcid.org/0000-0002-9264-4502
Joshua H Singer http://orcid.org/0000-0002-0561-2247
Wei Li http://orcid.org/0000-0002-2897-649X
Christophe P Ribelayga http://orcid.org/0000-0001-5889-2070
Stephen C Massey http://orcid.org/0000-0003-0224-6031

## Ethics

All animal procedures were reviewed and approved by the Animal Welfare Committee at the University of Texas Health Science Center at Houston (AWC-20-0138) or by our collaborators' local Institutional Animal Care and Use Committees.

## Decision letter and Author response

Decision letter https://doi.org/10.7554/eLife.73039.sa1
Author response https://doi.org/10.7554/eLife.73039.sa2

# Additional files

## Supplementary files

• Transparent reporting form

## Data availability

All the data used to create the figures in the manuscript have been provided as source data files for Figures 2, 3, 4, 5 and 8. The analyzed previously published SBF-SEM dataset e2006 is publicly available here: http://neuro.rzg.mpg.de/.

The following datasets were generated:

| Author(s) | Year | Dataset title | Dataset URL | Database and Identifier |
|---|---|---|---|---|
| Ishibashi M, Keung J, Ribelayga CP, Massey SC | 2018 | Confocal imaging of the outer plexiform layer in mouse retina | https://download.brainimagelibrary.org/30/67/30675648bee2309e/ | brainimagelibrary, 30675648bee2309e |
| Singer JH | 2018 | SBF-SEM of mouse retina. eel001 | https://wklink.org/9712 | webKnossos, 9712 |
| Morgan CW, Aicher SA, Carroll JR | 2019 | FIB-SEM of the outer plexiform layer in light-adapted mouse retina. EM1 and EM2 | https://bossdb.org/project/ishibashi2021 | BossDB, ishibashi2021 |

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

# Appendix 1

## Comparison of photoreceptor gap junctions

For comparison, we have summarized the evidence for rod/cone, cone/cone, and rod/rod gap junctions in the mouse retina.

### Rod/cone gap junctions

1. Cones contact all rods.
2. Cx36 labeling at the base of each rod spherule; no other connexins expressed by photoreceptors (*Jin et al., 2020*).
3. Cx36 labeling dramatically reduced in either rod- or cone-specific Cx36 KOs, consistent with the predominance of rod/cone gap junctions and the requirement for Cx36 on both sides of a rod/cone gap junction (*Jin et al., 2020*).
4. FIB-SEM shows gap junctions at the base of each rod spherule, coincident with cone contacts and Cx36 immunoreactivity.
5. Paired recordings show a large conductance between rod/cone pairs (resting state, dark-adapted), mean 300 pS (*Jin et al., 2020*).
6. Rod/cone coupling abolished in either the rod- or cone-specific Cx36 KO (*Jin et al., 2020*).
7. Rod signals transmitted to cones via rod/cone gap junctions (*Jin et al., 2020*).
8. Rod signals transmitted to ganglion cells via the secondary rod pathway are eliminated in either the rod- or cone-specific Cx36 KO, consistent with the predominance of rod/cone gap junctions and the requirement for Cx36 on both sides of a rod/cone gap junction (*Jin et al., 2022*).
9. Rod/cone gap junctions are common in several mammalian species (*O'Brien et al., 2012*; *Smith et al., 1986*).

We conclude that all rods are coupled to cones.

## Cone/cone gap junctions

1. Some cone/cone contacts, mostly between telodendria.
2. While Cx36 is colocalized with the cone telodendrial network, most Cx36 clusters are rod/cone gap junctions. We were unable to identify cone/cone Cx36 clusters in the mouse retina.
3. FIB-SEM shows some contacts between adjacent cones but no membrane density at contact points.
4. Paired recordings show weak cone/cone coupling, a fraction of rod/cone coupling, ~60 pS in mouse (*Jin et al., 2020*), but ~250 pS in ground squirrel where Cx36 between cones is prominent (*DeVries et al., 2002*).
5. Cone/cone coupling was preserved in the rod-specific Cx36 KO, suggesting cone/cone coupling is direct.
6. In the rod-specific KO, which should reveal cone/cone coupling, a few Cx36 clusters left, significantly different from cone-specific Cx36 KO or pan Cx36 KO (*Jin et al., 2020*).
7. Cone/cone coupling reported in other species, such as ground squirrel and primate.
8. In the rod-less, cone-only mouse, there is a large increase in Cx36 labeling in the OPL (*Dang et al., 2004*).

The evidence for cone/cone coupling is weak for the mouse retina. Despite the fact that we were unable to identify cone/cone gap junctions, there is physiological evidence for weak cone/cone coupling (*Jin et al., 2020*). While the presence of cone/cone coupling in the mouse is equivocal, there is strong evidence for cone/cone coupling in other species such as primate and ground squirrel (*DeVries et al., 2002*; *O'Brien et al., 2012*).

## Rod/rod

1. Few rod/rod contacts despite spherule packing in the OPL.
2. No Cx36 labeling at rod/rod contacts.

3. In the cone-specific Cx36 KO, which should reveal rod/rod coupling, there was no remaining Cx36 labeling in the OPL above a very low background (*Jin et al., 2020*).
4. FIB-SEM of rod/rod contacts shows no dense staining typical of a gap junction.
5. Paired recordings show apparent rod/rod coupling was abolished in the cone-specific Cx36 KO, indicating indirect or network coupling, rod/cone/rod (*Jin et al., 2020*).
6. Cx36 in the ONL is very weak, indistinguishable from nonspecific background labeling. If any immunolabeling represents gap junctions, as opposed to nonspecific noise, these very low numbers are not enough to account for coupling between the multitude of rods.

We conclude that there is no direct rod/rod coupling in the mouse retina.

In summary, the evidence for the predominant role of rod/cone gap junctions is strong and straightforward. In contrast, we found no evidence for rod/rod gap junctions and the distribution of Cx36 does not match the location of rod/rod contacts. We were also unable to detect cone/cone gap junctions by immunofluorescence or EM, but there is at least some supporting physiological evidence in favor of cone/cone gap junctions. It may seem unsatisfactory that we were unable to detect rod/rod or cone/cone gap junctions, but it does not change the main result: that rod/cone coupling accounts for the vast majority of photoreceptor gap junctions.

# Appendix 2

**Appendix 2—table 1.** Analysis of photoreceptors by serial blockface-scanning electron microscopy (SBF-SEM) from e2006.

|  | All cones | Central cones | p-Value |
|---|---|---|---|
| N of cones | 29 | 13 |  |
| N of rods | 811 | 361 |  |
| Rods/cone | 28.0 | 27.8 |  |
| Convergence |  |  |  |
| Mean ± SD | 43.0 ± 5.40 | 41.7 ± 4.19 | 0.40 |
| Median (Q1–Q3) | 43 (39–46) | 43 (38–45) |  |
| Divergence | 1.54 ± 0.628 | 1.89 ± 0.639 | $2.5 \times 10^{-17}$ |
| Conv./div. | 28.0 | 22.1 |  |
| Cone coverage area* |  |  |  |
| Mean ± SD | 104 ± 20.2 | 102 ± 14.5 |  |
| Median (Q1–Q3) | 100 (86.7–117) | 99.9 (93.0–111) |  |
| Area covered by cones* | 1920 | 890 |  |
| Cone density† | 0.0151 | 0.0146 |  |
| Coverage | 1.56 | 1.49 |  |
| N of shared rods |  |  |  |
| Mean ± SD | 6.23 ± 4.67 (N = 79) | 5.85 ± 4.42 (N = 61) | 0.63 |
| Median (Q1–Q3) | 4 (2–8) | 5 (2–9) |  |

Unit is *($\mu m^2$) and †(/$\mu m^2$).

**Appendix 2—table 2.** Comparison of blue and green cones.

|  | Blue cones | Green cones | p-Value |
|---|---|---|---|
| N of cones | 5 | 31 |  |
| Convergence |  |  |  |
| Mean ± SD | 39.8 ± 5.11 | 43.0 ± 6.86 | 0.31 |
| Median (Q1–Q3) | 39 (39–41) | 43 (39–46) |  |
| Cone coverage area* |  |  |  |
| Mean ± SD | 85.1 ± 18.6 | 104 ± 21.8 | 0.081 |
| Median (Q1–Q3) | 83.4 (76.4–98.0) | 105 (84.8–120) |  |
| N of shared rods |  |  |  |
| Mean ± SD | 6.56 ± 6.34 (N = 16) | 6.26 ± 4.14 (N = 69) | 0.86 |
| Median (Q1–Q3) | 6 (3–9) | 3.5 (2–10.8) |  |

Unit is *($\mu m^2$).

**Appendix 2—table 3.** Focused ion beam-scanning electron microscopy (FIB-SEM) analysis.

|  |  |
|---|---|
| N of rods | 42 |
| N of gap junction/rod |  |
| Mean ± SD | 3.21 ± 1.23 |

*Appendix 2—table 3 Continued on next page*

*Appendix 2—table 3 Continued*

| | |
|---|---|
| Median (Q1–Q3) | 3 (2–4) |
| Gap junction position* | |
| Mean ± SD | 0.686 ± 0.635 |
| Median (Q1–Q3) | 0.435 (0.345–0.703) |
| Synaptic mouth radius* | 0.138 ± 0.121 |
| Gap junction width* | |
| Mean ± SD | 0.123 ± 0.0320 |
| Median (Q1–Q3) | 0.120 (0.103–0.138) |
| Gap junction length* | |
| Mean ± SD | 0.477 ± 0.227 |
| Median (Q1–Q3) | 0.419 (0.332–0.568) |
| Total gap junction length/rod* | |
| Mean ± SD | 1.53 ± 0.439 |
| Median (Q1–Q3) | 1.52 (1.33–1.85) |

Unit is *(µm).

## Appendix 3

## Calculations of open probability for rod/cone gap junctions with cumulative errors

### Error propagation

The variability of our estimates of the fraction of opened channels under the different conditions tested (rest, w/quinpirole, w/spiperone) was calculated using the Taylor expansion method for the moments of functions of random variables and the Goodman formula (*). We used the means and respective SEMs of our data or of published data to calculate the mean and standard errors (SEs) of the estimates. A bootstrapping approach was necessary to derive means and SEMs from the nonparametrically distributed values of rod/cone transjunctional conductances (**).

> E1 – mean length of an FIB-SEM rod/cone gap junction: 477 ± 19.5 nm (n = 135, from *Appendix 2—table 3*)
> E2 – mean Cx36 channel-to-channel spacing: 9.45 nm ± 0.296 (***)

From E1 and E2, we calculated:

> E3 – mean number of Cx36 channels/rod/cone gap junction: $477/9.45 \pm \sqrt{[(19.5/9.45)^2 + (477/9.45)^2 \times (0.296/9.45)^2]} = 50.5 \pm \sqrt{(4.26 + 2548 \times 0.000981)} = 50.5 \pm 2.60$
> E4 – mean number of gap junctions/rod spherule: 3.21 ± 0.190 (n = 42, from *Appendix 2—table 3*)
> E5 – mean number of cone contacts/rod spherule (rod/cone divergence): 1.89 ± 0.0337 (n = 361, from *Appendix 2—table 3*)

From E4 and E5, we calculated:

> E6 – mean number of gap junctions between a rod/cone pair: $3.21/1.89 \pm \sqrt{[(0.190/1.89)^2 + (3.21/1.89)^2 \times (0.0337/1.89)^2]} = 1.70 \pm \sqrt{(0.0101 + 2.88 \times 0.000318)} = 1.70 \pm 0.105$

From E3 and E6, we calculated:

> E7 – mean number of Cx36 connexons between a rod/cone pair: $50.5 \times 1.70 \pm 50.5 \times 1.70 \times \sqrt{[(2.60^2/50.5^2 + 1) \times (0.105^2/1.70^2 + 1)] - 1} = 85.9 \pm \sqrt{[(0.00265 + 1) \times (0.00382 + 1)] - 1} = 85.9 \pm 6.91$
> E8 – mean unitary conductance of a Cx36 channel: 14.3 ± 0.8 pS (n = 92, from *Teubner et al., 2000*; number is in accordance with *Srinivas et al., 1999* [~15 pS])

From E7 and E8, we calculated:

> E9 – maximal conductance between a rod/cone pair if all Cx36 channels were open: $85.9 \times 14.3 \pm 85.9 \times 14.3 \times \sqrt{[(6.91^2/85.9^2 + 1) \times (0.8^2/14.3^2 + 1)] - 1} = 1228 \pm \sqrt{[(0.00647 + 1) \times (0.00313 + 1)] - 1} = 1228 \pm 120$ pS
> E10 – mean rod/cone conductance measured at rest (darkness): 307 pS ± 2.31 pS (n = 500, bootstrapping from data in *Jin et al., 2020*, **)

From E9 and E10, we calculated:

> E11 – mean fraction of open channels at rest (darkness): $307/1228 \pm \sqrt{[(2.31/1228)^2 + (307/1228)^2 \times (120/1228)^2]} = 0.250 \pm \sqrt{(0.00000354 + 0.0626 \times 0.00955)} = 0.250 \pm 0.0245 = 25.0\% \pm 2.45\%$
> E12 – mean rod/cone conductance measured in the presence of quinpirole: 35.5 ± 0.968 pS (n = 500, bootstrapping from data in *Jin et al., 2020*, **)

From E9 and E12, we calculated:

> E13 – mean fraction of open channels w/quinpirole: $35.5/1228 \pm \sqrt{[(0.968/1228)^2 + (35.5/1228)^2 \times (120/1228)^2]} = 0.0289 \pm \sqrt{(0.000000621 + 0.000836 \times 0.00955)} = 0.0289 \pm 0.00293 = 2.89\% \pm 0.293\%$
> E14 – mean rod/cone conductance measured in the presence of spiperone: 1312 ± 22.8 pS (n = 500, bootstrapping from data in *Jin et al., 2020*, **)

From E9 and E14, we calculated:

> E15 – mean fraction of open channels w/spiperone: $1312/1228 \pm \sqrt{[(22.8/1228)^2 + (1312/1228)^2 \times (120/1228)^2]} = 1.07 \pm \sqrt{(0.000345 + 1.14 \times 0.00955)} = 1.07 \pm 0.106 = 107\% \pm 10.6\%$

We can calculate 95% confidence intervals [lower limit, upper limit] (****) for E11, E13, and E15:

> E11 – mean open fraction (%) of open channels at rest (darkness): [20.2, 29.8]
> E13 – mean open fraction (%) of open channels w/quinpirole: [2.32, 3.46]
> E15 – mean open fraction (%) of open channels w/spiperone: [86.2, 128]

*Notes*:

**(*) Formulas to calculate SEs of our estimates**:

Taylor expansion method for the moments of functions of random variables (*Wolter, 1985*):

$E(X)$ or $E(Y)$ is the mean and $SEM(X)$ or $SEM(Y)$ is the standard error of the mean (SEM). Assuming that X and Y are random variables (independent), x and a are used to estimate $E(X)$ and $SEM(X)$, and y and b are for $E(Y)$ and $SEM(Y)$, we can calculate the SE of the ratio x/y according to the formula:

$SE(x/y) = \sqrt{(a/y)^2 + (x/y)^2 \times (b/y)^2}$

The SE of a product of wo independent variables can be calculated from a simplification of the Goodman formula (*Goodman, 1962*):

$SE(xy) = xy \times \sqrt{[(a^2/x^2 + 1) \times (b^2/y^2 + 1) - 1]}$

**(**) Bootstrapping**:

We used a classical method of bootstrapping (*Efron, 1979*; *Efron and Tibshirani, 1993*) to estimate the mean and SEM of nonparametric distributions of rod/cone gap junction conductances. Resampling with replacement from the original dataset was performed with a sample size of 2/3 of the original dataset (rounded to the closest whole number), 550 simulated datasets were generated, and the calculations were done on the last 500.

**(***) Estimation of Cx36 channel center-to-center spacing distance**:

One reasonable way to calculate the spacing distance in Cx36 channels is from the channel density of crystalline plaques, which consist of regular hexagonal arrays. From *Szoboszlay et al., 2016*, the maximum connexon packing density is $12,940 \pm 405$ channels/$\mu m^2$ (n = 12), a value in close agreement with other studies (see, for instance, *Kamasawa et al., 2006* [~12,000 channels/$\mu m^2$]). Assuming that the channels do not overlap, we can calculate a spacing value based on the formula for the area of a regular hexagon: channel density = $2/(\sqrt{3} \times r^2)$, where the channel density is the number of channels/$\mu m^2$ and r is the center-to-center spacing in $\mu m$.

Solving the equation for a density = 12,940 channels/$\mu m^2$ gives $r = 0.00945\ \mu m = 9.45$ nm, with SEM = $9.45 \times 405/12940 = 0.296$. Thus, mean center-to-center spacing of Cx36 channels in crystalline plaques is $9.45 \pm 0.296$ nm.

**(****) 95% CI of an estimate, such as ratio or product of means:**

[estimate - $1.96 \times SE$, estimate + $1.96 \times SE$]

