## [Editor Report]

This article presents a beautiful analysis of gap junctions in the outer retina using a combination of confocal imaging and electron microscopy. The result is a thorough description of connectivity between rod and cone photoreceptors, and a clear resolution of ambiguities present in past work on this topic.

---

## [Decision Letter]

**Decision letter after peer review:**

Thank you for submitting your article "Analysis of Rod/Cone Gap Junctions from the Reconstruction of Mouse Photoreceptor Terminals" for consideration by *eLife*. Your article has been reviewed by 3 peer reviewers, including Fred Rieke as the Reviewing Editor and Reviewer #1, and the evaluation has been overseen by Claude Desplan as the Senior Editor. The following individual involved in review of your submission has agreed to reveal their identity: Joshua Morgan (Reviewer #2).

The reviewers have discussed their reviews with one another, and the Reviewing Editor has drafted this to help you prepare a revised submission. The reviewers all appreciated the quality of the data in the paper, and felt that the main conclusions were well supported by the data. Two areas need improvement:

1. Additional quantification is needed for several conclusions (see reviewer #2 comments especially).

2. The motivation for the paper and relevance of the conclusions for an audience beyond the retina need to be strengthened considerably. One possible direction that came up in discussions among the reviewers was the technical strength of the combination of EM and confocal microscopy, and whether similar approaches could be used in other circuits.

*Reviewer #1 (Recommendations for the authors):*

Figures:

- Some identifying labels would be helpful. For example, pointing to a few telodendria in Figure 1 would be useful.

- Figure 1G looks like a different perspective from Figure 1D. It would help to clarify when you are showing things in cross section and when from above/below – perhaps with a little icon of a cone?

Correlating the figures and figure legends was difficult. Some of the figures (e.g. Figure 1) were cut off, and the figures did not have any identifying labels (so I had to count from the start each time I wanted to be sure I was looking at the right one). Ideally the legends would appear with the figure to ease reviewing!

Lines 77 and 99 make the same point

Lines 94-96 introduce rod-rod coupling, but then that comes up again in the next paragraph.

Line 134: should this refer to Figure 1C?

Line 200: very nice analysis!

Line 254: can you label cones in Figure so the reader knows which is cone 5?

Line 317: can you clarify what "merged" means here – as worded it sounds like they become one (like a synaptic vesicle merging or fusing with the cell membrane).

*Reviewer #2 (Recommendations for the authors):*

Line 85: "In lower vertebrates" is outdated terminology ("In non-mammalian vertebrates"?) and may also be overly broad for the reference ( "In salamanders.."?).

Figure 1: Please include visible background in all fluorescence images. Images are filtered. is filtering described?

Figure 1D: The red channel is masked and binerized. It is not possible to judge the signal to background ratio or to determine if the red spots are distinct puncta or the largest parts of a cytosolic fill. Please provide an unsaturated version of this image. Please also list image manipulations like masking in the figure legend

Figure 2: The imaging and analysis here are great. However, the images also look heavily processed, especially the ribeye. Please describe the processing. If possible, provide a supplement of raw images where background signal, noise, unprocessed shapes are visible.

Line 306: You absolutely cannot say things like "X is smaller than Y, but the difference is not significant due to small sample size". Sample size is where you derive confidence to make scientific claims. In your sample, X was smaller than Y, but your sample size is too small to be confident that this difference represents the population of X and Y. You absolutely cannot say that you know what the answer would be if you sampled more.

Line 308: The indistinguishable sentence is too strong. By all other measures Blue cones are indistinguishable…. That suggests a conclusion about the population for "all other" measures. What you have is a limited set of measures for 5 cones. It is accurate to say the 5 Blue cones we measured were within the range we observed for green cones in terms of convergence (Blue X+-X, Green X+-X), blank () and blank(). To make a strong statistical argument about similarity you would have to do something along the lines of a power test and your sample size is too small to make that meaningful.

Line 623: The EM evidence for the absence of Cone/Cone gap junctions in mice is weak. Only 6 cells from one patch of one retina were used. Mentioning the 20 examples of cone/cone contacts is useful but can be a little misleading without context. You can readily say that the vast majority of gap junctions in your tissue are cone/rod. However, you can't rule out the possibility of some cone/cone gap junctions in other photoreceptors in other parts of other retinas or that something in the prep made cone/cone gap junctions difficult to detect. Given that the prior prediction was that cone/cone would be much less than rod/cone I don't consider the results presented here to be far out of line of the expected. The result could certainly be strengthened by checking other retinas. Alternatively, a discussion of the limits of the small sample could be included in the Discussion section.

Line 714: There are no error bars on your estimate of 89% open. Given the number of inferences, it seems plausible that the estimate of channel number could be off by several fold. Any underestimate of gap junction length could push the open fraction to the " very surprising" large number. My understanding is that the 10nm center to center number is a single measure (no population numbers) from the IPL. That number is combined with the observation that single string gap junctions are found between rods and cones in primates.

Possible sources of error

– Variation in population center to center spacing

– Doubling in channel packing

– Tissue shrinkage from aldehyde fixation and solvent dehydration vs frozen tissue.

– Failure to detect some gap junctions

– Species differences

– Multiple chains interpreted as single chains.

Possible Circadian Issue: I don't believe any mention is made of the time of day the animals were sacrificed or whether circadian regulation of coupling might have a structural component. If possible, provide this data or include in discussion.

*Reviewer #3 (Recommendations for the authors):*

– Abstract: I think you could make the abstract more engaging by adding a motivational sentence or two and a strong summary sentence with the main results

– Overall manuscript: As presented here, the manuscript reads like a concatenation of datasets and their analysis which would benefit from more emphasis on a common thread throughout the sections

– The end of the discussion is quite abrupt. A closing paragraph that sums up the results is missing

– As it currently is, I'm missing a discussion of the implications of your finding for the field and possible relevance for future research or emerging research questions

– The source data is not really source data, merely the counts in excel files. From an open-source point of view, it would be nice to at least have it as csv file so that everyone can open it including via python etc.

– L. 61: I don't see and know any evidence that rod spherules are structured in layers

– Figure 1 supp 2C: I can't see any cyan labeling, only magenta next to some green labeling, is it that what you mean? (L. 1282)

– Figure 3: What do the black arrows mean, it is nowhere mentioned what they should highlight.

– L. 493: Something like "Higher resolution FIB-SEM of a smaller volume.…" would be clearer

– L. 496: Where does that conductance value come from? This is/could be one of the more important statements of the paper and it should appear in the Results section. I suggest moving ll. 700 further up so that the calculation comes earlier in the manuscript.

– Ll. 919: I'm not sure I understand how you measured the length of the GJ as a rectangular shape. How big is the deviation from the true length given a structure that looks more like three quarters of a circle as shown in Figure 7G,H? Could you clarify this with a sketch of how you measured gap junction size?

[Editors’ note: further revisions were suggested prior to acceptance, as described below.]

Thank you for resubmitting your work entitled "Analysis of Rod/Cone Gap Junctions from the Reconstruction of Mouse Photoreceptor Terminals" for further consideration by eLife. Your revised article has been evaluated by Claude Desplan (Senior Editor) and a Reviewing Editor.

The manuscript has been improved but there is a remaining suggestion that we would like you to consider: You need to rewrite the abstract, introduction and discussion in order to much better integrate the new results/text rather than patching them onto the previous version. These editorial changes, which could be done in a few days, would allow the readers to much better understand the significance of the paper.

Reviewer #1 (Recommendations for the authors):

This paper has improved in revision, but more work could be done to integrate the changes into the paper. The text added to the Introduction does a better job highlighting the importance of gap junctions, but the introduction now has two parts that could be integrated much more smoothly. For example, it could be clearer which outer retina gap junctions could contribute to signal averaging and which to a separate pathway. Related to this point, there is some repetition between the first few (new) paragraphs of the introduction and the text about the outer plexiform layer.

Related to this point, the Discussion is still relatively focused on the retina. I think it would strengthen the paper if you can come back to the general issues that are not described early in the introduction and summarize your findings in the context of those issues.

---

## [Author Response]

The reviewers have discussed their reviews with one another, and the Reviewing Editor has drafted this to help you prepare a revised submission. The reviewers all appreciated the quality of the data in the paper, and felt that the main conclusions were well supported by the data. Two areas need improvement:1. Additional quantification is needed for several conclusions (see reviewer #2 comments especially).

We have addressed several quantification issues:

1. Reviewer 2 correctly points out that the data concerning blue cone pedicles was not significant and it is inappropriate to draw conclusions based on this limited data. The text has been revised to correct this issue.

2. In the FIB-SEM data, the number of cones may be insufficient to draw firm conclusions. The text has been changed to soften this conclusion.

3. Reviewer 2 requested that we include the potential error in calculating the gap junction open probability values. We have calculated the cumulative error in deriving the open probability: errors are now included in the text along with the numerical data, and the calculation methods have been added in Appendix 3. As requested by Reviewer 2, we have included a section in the discussion considering possible sources of error in our calculations. This is now presented in a transparent manner.

2. The motivation for the paper and relevance of the conclusions for an audience beyond the retina need to be strengthened considerably. One possible direction that came up in discussions among the reviewers was the technical strength of the combination of EM and confocal microscopy, and whether similar approaches could be used in other circuits.

We have included additional material covering the motivation and a broader background to the introduction and discussion, as requested by Reviewers 1 and 3. In the discussion, we have added text concerning the combination of confocal microscopy and EM, as suggested. We have also added a final section to the discussion, as requested by reviewer 3.

Reviewer #1 (Recommendations for the authors):Figures:- Some identifying labels would be helpful. For example, pointing to a few telodendria in Figure 1 would be useful.

It was necessary to split the previous Figure 1 into two parts. We have taken the opportunity to enlarge Figure 1A and placed some arrows pointing to cone telodendria, as requested.

- Figure 1G looks like a different perspective from Figure 1D. It would help to clarify when you are showing things in crossection and when from above/below – perhaps with a little icon of a cone?

We couldn’t think of an intuitively obvious way to use a cone icon. Instead, we have specifically labeled the former Figure 1D (now in Figure 2) as a 3D projection, because this image is the exception and seems to have caused the most trouble. In addition, we have added new 2D figures from this confocal series to show how the projection of the cone and colocalized Cx36 clusters was developed.

Correlating the figures and figure legends was difficult. Some of the figures (e.g. Figure 1) were cut off, and the figures did not have any identifying labels (so I had to count from the start each time I wanted to be sure I was looking at the right one). Ideally the legends would appear with the figure to ease reviewing!

We apologize for the lack of figure numbers; this has now been corrected.

Lines 77 and 99 make the same point

The phrase on line 99 was deleted.

Lines 94-96 introduce rod-rod coupling, but then that comes up again in the next paragraph.

Text on rod/rod coupling has been amalgamated in one paragraph, as below.

Revised text:

“Rods and cones both express Cx36, but no other connexins (Bloomfield and Völgyi, 2009; Jin et al., 2020; O’Brien et al., 2012), and, in fact, Cx36 mediated rod/cone coupling accounts for most of the gap junctions in the OPL (Asteriti et al., 2017; Ingram et al., 2019; Jin et al., 2020; Miller et al., 2017). If indeed rods express Cx36, then it raises the possibility that they could also be coupled via Cx36 gap junctions. Rod/rod gap junctions have been suggested from tracer coupling studies and from EM studies of the mouse retina (Jin et al., 2015; Li et al., 2012; Tsukamoto et al., 2001), and, in salamander retina, rods are extensively electrically coupled (Zhang and Wu, 2004). However, the evidence for rod/rod gap junctions in the mammalian retina is mixed (Bloomfield and Völgyi, 2009; Jin et al., 2020; Tsukamoto et al., 2001) and will be addressed here.”

Line 134: should this refer to Figure 1C?

Thank you, yes, it should be former 1C, now part of Figure 2. Changed.

Line 200: very nice analysis!

Thank you!

Line 254: can you label cones in Figure so the reader knows which is cone 5?

A bold green outline has been added to show cone 5 in Figure 5B and C. Cone 2 (blue) and cone 3 (cyan) are also outlined.

Line 317: can you clarify what "merged" means here – as worded it sounds like they become one (like a synaptic vesicle merging or fusing with the cell membrane).

Done,

“The contact sites, where the cell membrane of a cone telodendron combined with the cell membrane of a rod spherule, with no visible space between them, were highlighted.”

Reviewer #2 (Recommendations for the authors):Line 85: "In lower vertebrates" is outdated terminology ("In non-mammalian vertebrates"?) and may also be overly broad for the reference ( "In salamanders.."?).

Changed.

Figure 1: Please include visible background in all fluorescence images. Images are filtered. is filtering described?

No filtering was used for any of the confocal images. As requested, we have revised the brightness/contrast of the images to provide a little background to enable the reader to judge the specificity of the immunolabeling.

Figure 1D: The red channel is masked and binerized. It is not possible to judge the signal to background ratio or to determine if the red spots are distinct puncta or the largest parts of a cytosolic fill. Please provide an unsaturated version of this image. Please also list image manipulations like masking in the figure legend

The former Figure 1D, which shows a 3D projection of a single EGFP labeled cone with colocalized Cx36, now part of Figure 2, has been revised. The first panels, 2B and C, show the original 2D sections to demonstrate the distribution of Cx36 clusters, restricted to the cone pedicles, stained for cone arrestin, as well as the EGFP labelled cone. Background was added, as requested. In Figure 2C, only the Cx36 that was colocalized with the EGFP cone was retained but we hope the procedure is now clear. The 3D projections of the single EGFP labeled cone were made from these 2D sections and these panels are now labeled as 3D projections and Cx36 is labeled as colocalized for clarity. In addition, raw single channel images have been added as supplemental Figures. The procedure is described in the figure legend and the text has been revised.

Figure 2: The imaging and analysis here are great. However, the images also look heavily processed, especially the ribeye. Please describe the processing. If possible, provide a supplement of raw images where background signal, noise, unprocessed shapes are visible.

Thank you, we are pleased you like the imaging and analysis.

Former Figure 2, now Figure 3, has no processing or filtering except brightness and contrast. It has been adjusted to provide background. Original single channel images showing background are in a supplement (Figure 2, suppl. 1).

Line 306: You absolutely cannot say things like "X is smaller than Y, but the difference is not significant due to small sample size". Sample size is where you derive confidence to make scientific claims. In your sample, X was smaller than Y, but your sample size is too small to be confident that this difference represents the population of X and Y. You absolutely cannot say that you know what the answer would be if you sampled more.

Thank you. This is quite correct, and we have revised the text.

Line 308: The indistinguishable sentence is too strong. By all other measures Blue cones are indistinguishable…. That suggests a conclusion about the population for "all other" measures. What you have is a limited set of measures for 5 cones. It is accurate to say the 5 Blue cones we measured were within the range we observed for green cones in terms of convergence (Blue X+-X, Green X+-X), blank () and blank(). To make a strong statistical argument about similarity you would have to do something along the lines of a power test and your sample size is too small to make that meaningful.

We have revised the text, removing indistinguishable, as below. We have added that the blue cone contacts with rods fall within the range of green cones.

“Blue Cone Skeletons also Contact Rod Spherules

The skeleton of a blue cone, defined by its bipolar cell contacts (Behrens et al., 2016; Nadal-Nicolás et al., 2020), is also shown in figure 5A and the analysis of all five blue cones from the dataset of Behrens et al., (2016) is presented for comparison with green cones (Figure 5F and G). The blue cone data are summarized in Table 2, Appendix 2. The telodendrial area of blue cones was, 85.1 ± 18.6 μm^2^ (mean + SD, n=5) (Figure 5F). Like green cones, blue cones contact all rod spherules, within their telodendrial field (convergence 39.8 + 5.11, mean + SD, n=5). The convergence in blue cones fell within the range of rods in contact with green cones, 43.0 + 5.40 rod spherules (mean + SD, n=29). Most of the rods in contact with a blue cone also receive contact from adjacent overlapping green cones. Thus, there is no evidence for color selective rod contacts.”

Line 623: The EM evidence for the absence of Cone/Cone gap junctions in mice is weak. Only 6 cells from one patch of one retina were used. Mentioning the 20 examples of cone/cone contacts is useful but can be a little misleading without context. You can readily say that the vast majority of gap junctions in your tissue are cone/rod. However, you can't rule out the possibility of some cone/cone gap junctions in other photoreceptors in other parts of other retinas or that something in the prep made cone/cone gap junctions difficult to detect. Given that the prior prediction was that cone/cone would be much less than rod/cone I don't consider the results presented here to be far out of line of the expected. The result could certainly be strengthened by checking other retinas. Alternatively, a discussion of the limits of the small sample could be included in the Discussion section.

We agree that the sample of 6 cones is too small to definitively rule out cone/cone coupling in the mouse. We have revised the text to make this explicitly clear (see below). Indeed, the evidence from our previous work suggests there is a small amount of cone/cone coupling, despite the fact that we could not find cone/cone gap junctions. We have also added a phrase to make it clear that the 20 cone/cone contacts that we report without the characteristics of gap junctions were derived from these 6 partially reconstructed cone pedicles in one FIB-SEM dataset.

Revised text:

“Cone/Cone Contacts were not Detected

We were unable to confirm the presence of cone/cone gap junctions in the mouse retina, despite locating more than 20 examples of cone/cone contacts from six partially reconstructed cone pedicles in one FIB-SEM dataset……….

The available data was derived from a limited set of six partially reconstructed cones, due to the high-resolution, yet small area of the FIB-SEM dataset. Thus, while we are confident that rod/cone coupling accounts for the vast majority of photoreceptor gap junctions, the sample size is too small to rule out a minor amount of cone/cone coupling, perhaps too small or too faintly stained to be readily detected. Our previous electrophysiological studies suggest there is weak cone to cone coupling in the mouse retina, which……..”

Line 714: There are no error bars on your estimate of 89% open. Given the number of inferences, it seems plausible that the estimate of channel number could be off by several fold. Any underestimate of gap junction length could push the open fraction to the " very surprising" large number. My understanding is that the 10nm center to center number is a single measure (no population numbers) from the IPL. That number is combined with the observation that single string gap junctions are found between rods and cones in primates.Possible sources of error– Variation in population center to center spacing– Doubling in channel packing– Tissue shrinkage from aldehyde fixation and solvent dehydration vs frozen tissue.– Failure to detect some gap junctions– Species differences– Multiple chains interpreted as single chains.

This is a major issue. Therefore, we have calculated the cumulative errors in presenting the open probability of rod/cone gap junction channels, which is a ratio of the mean conductance under different conditions (from our previously published physiological work Jin et al., 2020), over the maximum potential conductance, derived from the morphological data in this paper, interpreted with reference to earlier work on photoreceptor gap junctions. The new numbers are slightly different from the previous calculations because we used the means rather than the medians in our calculations, consistent with statistical practice. However, it should be noted that they are not qualitatively different. All numbers for open probability in the text are now expressed as a percentage with the SEM. Appendix 3 showing our calculations is now attached.

We recognize that there are several assumptions and potential sources of error in these calculations and we agree that it is appropriate to disclose them. Therefore, in the interests of transparency, we have added a section to the discussion detailing possible sources of error. In this text, we have addressed all issues raised by reviewer 2.

Added to Discussion:

“Possible Sources of Error

It is important to note that there are several assumptions and potential sources of error which may affect these calculations. The structure of small gap junctions lies below the resolution of conventional light microscopy and must be inferred. Likewise, in EM material, the identity of the darkly stained chromophilic material is unknown and may include auxiliary proteins in addition to connexins which may exaggerate the dimensions of a gap junction, particularly in width. First, our calculations are based on the length of gap junction strings with a width of 1 connexon because the freeze-fracture studies show a string of single gap junction particles on the surface of a rod spherule (Raviola and Gilula, 1976; Reale et al., 1978). This is consistent with the low intensity of Cx36 labeling in the OPL, which indicates the presence of very small or sparse gap junctions such as strings (Dang et al., 2004; Mills et al., 2001). If in fact the string width were two connexons, it would double the channel number and thus halve the open probability when compared with our previous conductance measurements from paired recordings (Jin et al., 2020). Double strings, two particles wide were occasionally observed but they were not the predominant form (Raviola and Gilula, 1976). Both freeze-fracture studies were from primate, one macaque and one human, so it is also possible there is a species difference for the mouse, which could perhaps be addressed in the future by super resolution microscopy.

Secondly, species variation in the packing density of gap junction channels may also affect our calculations. Here, we are on safer ground. The connexon channel density was derived from two freeze fracture studies in mouse which are in good agreement with a value around 12,000 connexons/μm^2^ for Cx36 plaques (Kamasawa et al., 2006; Szoboszlay et al., 2016). We simply calculated the center-to-center spacing based on these numbers (Appendix 3) and it closely agrees with the 10 nm particle spacing of gap junction strings in photoreceptors (Raviola and Gilula, 1976). A third source of error could be that we have failed to detect some rod/cone gap junctions artificially increasing the open probability numbers. However, this is probably a minor issue because we were able to detect gap junctions with a minimum length of approximately 100nm, or ~ 10 connexons (Figure 8). Fourthly, we have made no attempt to correct our numbers for shrinkage due to tissue fixation or to compensate for the specific processing required by different techniques. Some measurements of gap junction position relied on both confocal microscopy and SEM, and these techniques gave comparable results (Figures 3E and 8B, C). Furthermore, both freeze fracture and FIB-SEM gave similar results for rod/cone gap junction length, ~500nm, albeit from different species, primate and mouse respectively. Therefore, the errors introduced by the use of these different techniques are not large and do not materially affect our conclusions.

Finally, previous work reported that electrical coupling between photoreceptors changes with the time of day, reflecting the influence of light/dark adaptation and circadian clocks (Ribelayga et al., 2008; Li et al., 2009; Li et al., 2013; Zhang et al., 2015; Jin et al., 2015; Jin et al., 2016; Jin et al., 2020). In addition, circadian changes in photoreceptor coupling are found in mouse lines that are proficient in melatonin synthesis, such as the CBA/CaJ line (Jin et al., 2016). Yet, daily changes in photoreceptor coupling reflect changes in Cx36 phosphorylation state and open probability, without significant structural changes in the number or size of Cx36 gap junctions present (Li et al., 2009; Li et al., 2013; Zhang et al., 2015). Thus, circadian changes are not likely to affect our conclusions concerning the structure of rod/cone gap junctions.

While we have listed these potential sources of error in the interest of transparency, we note that a combined under-estimation of the connexon number by a factor as large as two yields a maximum open probability number of >50% (107%/2). This would still provide a historically high number for the open probability of gap junctions. Thus, we believe our calculations predict that the fraction of connexons which participate in gap junction modulation is greater than previously appreciated. Our results suggest that modulation of gap junction strings may include most connexon channels, perhaps approaching 100% when pharmacologically driven to an extreme.”

Possible Circadian Issue: I don't believe any mention is made of the time of day the animals were sacrificed or whether circadian regulation of coupling might have a structural component. If possible, provide this data or include in discussion.

We do not believe that circadian changes are reflected in gap junction structural changes. This is now included in the discussion of potential sources of error (see above). In addition, a statement has been added to the Methods section noting that animals were euthanized at the same time of day and under the same conditions.

Reviewer #3 (Recommendations for the authors):– Abstract: I think you could make the abstract more engaging by adding a motivational sentence or two and a strong summary sentence with the main results

We have added new material to the abstract and introduction explaining the reason for the study. Reviewer 1 also requested more general background for the introduction.

“Electrical coupling, mediated by intercellular pores known as gap junctions, contributes to signal averaging, synchronization and noise reduction in neuronal circuits. However, because gap junctions are small and difficult to stain, they are often ignored in large-scale datasets used for 3D reconstruction. Here, we show an example of small-scale connectomics to measure gap junctions between photoreceptors in the mouse retina, using serial blockface-scanning electron microscopy (SBF-SEM), focused ion beam-scanning electron microscopy (FIB-SEM), and confocal microscopy for the gap junction protein Cx36…”

– Overall manuscript: As presented here, the manuscript reads like a concatenation of datasets and their analysis which would benefit from more emphasis on a common thread throughout the sections– The end of the discussion is quite abrupt. A closing paragraph that sums up the results is missing– As it currently is, I'm missing a discussion of the implications of your finding for the field and possible relevance for future research or emerging research questions

As also requested by Reviewer 2, we have revised the methodological sub-headings to include the biological relevance.

As requested, we have added a closing section summing up and considering the implications for future work.

Additional Discussion:

“Summary and Future Directions

We have resolved the identity of gap junctions between photoreceptors. We have shown that rod/cone coupling is the predominant form of photoreceptor coupling in the mouse retina. In contrast to the primate and ground squirrel, there is little or no cone coupling in the mouse retina. The use of confocal microscopy to locate Cx36 gap junctions, combined with SBF-SEM and FIB-SEM was extremely valuable because we knew where to look to find gap junctions in the SEM material. This was aided in part by the simple and stereotyped morphology of the OPL. A similar approach could be used in other systems but if the gap junctions are distributed over a dendritic tree or along an axon, as opposed to a small rod terminal, they may be more difficult to find and image. It will be important to extend this approach to the inner retina, where the circuitry is more complex, and the number of neuronal types is much greater. More generally, our study demonstrates the utility of targeted, small-scale ‘connectomic’ analysis for the identification of neural circuit components.

However, all serial EM datasets are not equal. Some have reported the signature of gap junctions, notably for RC1, a transmission EM (TEM) dataset from rabbit retina (Marc et al., 2018; Sigulinsky, et al., 2020). TEM also has the advantage that sections can be resampled at higher resolution to confirm the pentalaminar structure of gap junctions, though this is impractical for every example. However, in other large datasets gap junctions are missing and their potential contributions have been overlooked (Kasthuri et al., 2015; Scheffer et al., 2020). The promise of connectomics, to decode neuronal circuits and function, cannot be fulfilled without the inclusion of gap junctions and electrical coupling (Scheffer and Meinertzhagen, 2021).

The calculations of connexon number and open probability, based on the combination of morphological data, cellular geometry and previous conductance measurements were revealing. Rod/cone gap junctions form a switchable pathway, known as the secondary rod pathway, controlled via D2 dopamine receptors with a mean of approximately 80 gap junction channels and a mean maximal conductance of ~1200pS per rod/cone pair. We estimate that a large proportion of connexon channels in these small string-like gap junctions may be active rather than passive. As we learn their basic properties, it may be possible to understand the contribution of gap junctions as circuit elements in specific neuronal pathways.”

– The source data is not really source data, merely the counts in excel files. From an open-source point of view, it would be nice to at least have it as csv file so that everyone can open it including via python etc.

We will provide all confocal and EM image files in publicly accessible web servers as original source data. In addition we provide the skeleton and cell-cell contacts data file of Knossos, Figure 3 data. Most of *eLife* papers provide source data in excel format to make it in visually organized style. Several programing platform can open excel file as Python does easily using Pandas. I believe excel format is fine for broad readers.

– L. 61: I don't see and know any evidence that rod spherules are structured in layers

Rewritten to clarify.

Revised Text:

“…the smaller rod spherules are found above and between the cone pedicles…”

– Figure 1 supp 2C: I can't see any cyan labeling, only magenta next to some green labeling, is it that what you mean? (L. 1282)

Thank you for catching this error. Figure legend changed to magenta to match the figure.

– Figure 3: What do the black arrows mean, it is nowhere mentioned what they should highlight.

Now Figure 4, the black arrow points to a single blue cone pedicle that is on the edge of the dataset. This is now stated in the figure legend.

– L. 493: Something like "Higher resolution FIB-SEM of a smaller volume.…" would be clearer

Changed, Higher resolution FIB-SEM (4nm voxels) of a smaller volume confirmed that gap junctions occur at.

– L. 496: Where does that conductance value come from? This is/could be one of the more important statements of the paper and it should appear in the Results section. I suggest moving ll. 700 further up so that the calculation comes earlier in the manuscript.

Thank you, we have rewritten this section to clarify that the mean theoretical maximum conductance was calculated from the morphological parameters measured in this paper, see below. We prefer to retain these calculations in the Discussion section because they rely on additional published data. The section is very clearly labeled as calculation in the sub-heading. However, we have added these numbers to the abstract and summary of the paper to emphasize the importance.

Revised Text:

“Multiplying the mean number of connexons by the single connexon channel conductance for Cx36 (~15pS), we calculated the mean theoretical maximum conductance, if all connexon channels were open, of ~1200pS per rod/cone pair, based on the morphological parameters alone.”

– Ll. 919: I'm not sure I understand how you measured the length of the GJ as a rectangular shape. How big is the deviation from the true length given a structure that looks more like three quarters of a circle as shown in Figure 7G,H? Could you clarify this with a sketch of how you measured gap junction size?

We have added a supplementary figure (Figure 7, suppl. 1) to clarify how gap junction length was measure, as requested.

[Editors’ note: what follows is the authors’ response to the second round of review.]

This paper has improved in revision, but more work could be done to integrate the changes into the paper. The text added to the Introduction does a better job highlighting the importance of gap junctions, but the introduction now has two parts that could be integrated much more smoothly. For example, it could be clearer which outer retina gap junctions could contribute to signal averaging and which to a separate pathway. Related to this point, there is some repetition between the first few (new) paragraphs of the introduction and the text about the outer plexiform layer.Related to this point, the Discussion is still relatively focused on the retina. I think it would strengthen the paper if you can come back to the general issues that are not described early in the introduction and summarize your findings in the context of those issues.

We have revised our paper as suggested by the senior editor and Reviewer 1. Specifically, we have merged the two different sections of the Introduction into one, with the general background concerning gap junctions and the motivation for the study coming first.

The issue about homologous/heterologous coupling between photoreceptors is now specifically addressed, lines 80 – 87.

We have also revised the Discussion to place greater emphasis on general issues which may be of interest to a wider audience. Thus, the broader discussion concerning gap junctions in serial EM datasets now comes first.